# MVGS: Multi-view-regulated Gaussian Splatting for Novel View Synthesis

## Abstract

Recent works in volume rendering, *e.g.* NeRF and 3D Gaussian Splatting (3DGS), significantly advance the rendering quality and efficiency with the help of the learned implicit neural radiance field or 3D Gaussians. Rendering on top of an explicit representation, the vanilla 3DGS and its variants deliver real-time efficiency by optimizing the parametric model with single-view supervision per iteration during training which is adopted from NeRF. Consequently, certain views are overfitted, leading to unsatisfying appearance in novel-view synthesis and imprecise 3D geometries. To solve aforementioned problems, we propose a new 3DGS optimization method embodying four key novel contributions: 1) We transform the conventional single-view training paradigm into a multi-view training strategy. With our proposed multi-view regulation, 3D Gaussian attributes are further optimized without overfitting certain training views. As a general solution, we improve the overall accuracy in a variety of scenarios and different Gaussian variants. 2) Inspired by the benefit introduced by additional views, we further propose a cross-intrinsic guidance scheme, leading to a coarse-to-fine training procedure concerning different resolutions. 3) Built on top of our multi-view regulated training, we further propose a cross-ray densification strategy, densifying more Gaussian kernels in the ray-intersect regions from a selection of views. 4) By further investigating the densification strategy, we found that the effect of densification should be enhanced when certain views are distinct dramatically. As a solution, we propose a novel multi-view augmented densification strategy, where 3D Gaussians are encouraged to get densified to a sufficient number accordingly, resulting in improved reconstruction accuracy. We conduct extensive experiments to demonstrate that our proposed method is capable of improving novel view synthesis of the Gaussian-based explicit representation methods about 1 dB PSNR for various tasks. Codes are available.

## 1 Introduction

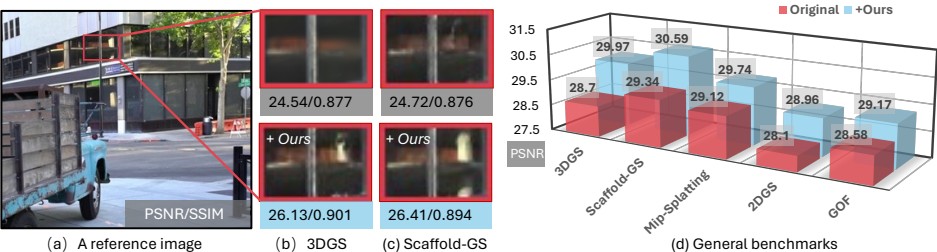

(a) A reference image     (b) 3DGS     (c) Scaffold-GS     (d) General benchmarks

Figure 1: **MVGS** supplements general improvements for novel view synthesis on top of GaussianSplatting (Kerbl et al., 2023) representations, as shown in (b) and (c). Extensive experiments are conducted to prove that our proposed method delivers consistent advantages in (d) in extremely challenging scenes with strong reflection, transparency, and fine-scale details against baseline methods.

Photorealistic rendering towards unbounded scenes or single object is proven to hold significant values in both the industrial and academic areas, *e.g.* multi-media generation, virtual reality, and autonomous driving. Conventional primitive-based representations such as mesh and point cloud (Botsch et al., 2005; Lassner & Zollhofer, 2021; Yifan et al., 2019; Munkberg et al., 2022) make real-time rendering

possible with the help of efficient rasterization. Although such a rendering mechanism delivers high efficiency, revealing a fine-grained and sufficiently precise appearance is still struggling, where blurry artifacts and discontinuity happen. On the contrary, implicit representation (Erler et al., 2020) and neural radiance field (Mildenhall et al., 2021; Barron et al., 2022; Müller et al., 2022)(NeRF), employ the multi-layer perceptron (MLP) to improve the potentiality of rendering high-fidelity geometries where more detailed structures are preserved. However, their inference efficiency is still limited even with accelerating operators such as Instant-NGP (Müller et al., 2022).

Recently, 3D Gaussian-based explicit representation, *e.g.* Gaussian Splatting (3DGS) (Kerbl et al., 2023; Jiang et al., 2024; Lu et al., 2024; Wu et al., 2024) achieve both state-of-the-art rendering quality and efficiency contributed by its tailored rasterization technique, following the paradigm of NeRF, *i.e.* training with a sample from a single camera view per iteration. Such a training strategy is commonly adopted in NeRF because of its pixel-wise rendering behavior for the convenience of utilization of supervision. As for 3DGS (Kerbl et al., 2023), 3D Gaussians kernels are directly rasterized on the image plane and get optimized with pixel-wise losses compared to the ground truth as well. However, because of the explicit characteristic of 3DGS representation, we observe the single-view training paradigm encourages 3D Gaussian kernels overfitting certain views for reducing training losses, making it not robust enough to precisely present all details in the scene.

In this paper, we propose *MVGS*, a general optimization method, empowering a large variety of Gaussian-based explicit approaches for better NVS precision as shown in Fig. 1 (d), with an exemplar case as shown in Fig. 1 (a), (b) and (c). The most crucial contribution of our work is altering the traditional training paradigm using a single-view supervision per iteration. We propose to incorporate multiple views per iteration during training by the proposed multi-view-regulated learning. Specifically, the overall set of 3D Gaussians towards the scene is forced to learn the structure and appearance of multiple views jointly without suffering the overfitting issues from a specific view. Consequently, such an optimization enables 3DGS kernels to get constrained to satisfy the rendering for a selection of views instead of overfitting to a certain view. To incorporate more information upon multi-view supervision, we propose cross-intrinsic guidance from low resolution to high resolution during training. The low-resolution training allows plenty of multi-view information as a powerful constraint to build more compact 3D Gaussians, which also conveys learned scene structure for high-resolution training to sculpt finer detail. Intuitively, the 3D Gaussians in overlapped 3D regions of cross rays should be densified to improve reconstruction performance for these views since these 3D Gaussians jointly serve and play an important role in the rendering of these views. To foster the effectiveness of learning multi-view information, we further propose a cross-ray densification strategy to guide the densification process, utilizing the ray-marching techniques with the guidance of the 2D loss maps. In addition, we propose a multi-view augmented densification strategy when discrepancies between perspectives are significant. This approach encourages 3D Gaussian to densify more primitives, enabling better fitting across various perspectives and improving performance.

Extensive experiments are conducted to demonstrate that our method effectively improves NVS performance for state-of-the-art Gaussian-based methods on various tasks, including general and reflective object reconstruction, 4D reconstruction, and large-scale scene reconstruction. Particularly, our experiments indicate that the NVS precision improves as the number of views increases in each optimization round. Moreover, our method encourages the learned 3D Gaussians to be more compact for representing the entire scene due to our proposed multi-view regulated learning. In conclusion, we summarize our contributions as below:

• We first propose a multi-view regulated training strategy that can be easily adapted upon existing single-view supervised 3DGS framework and its variants optimized for a large variety of tasks, where the NVS and normal precision can be consistently improved.

• Inspired by the benefit introduced by multi-view supervision with different extrinsic setups, a cross-intrinsic guidance scheme is proposed to train 3D Gaussians in a coarse-to-fine way. So that 3D Gaussians can accommodate higher consistency with pixel-wise local features.

• As densification strategy is crucial for 3DGS, we further propose a cross-ray densification strategy, emitting rays under 2D loss map guidance and densifying for overlapped 3D regions. The densified 3D Gaussians in those overlapped regions facilitate the fitting of multiple views, improving the performance of novel view synthesis.

• Last but not least, we propose a multi-view augmented densification strategy, intensifying densification while the discrepancies of multiple views are significant. It ensures that 3D Gaussians can be densified sufficiently to fit well with dramatically changed multi-view supervised information.

• In summary, extensive experiments demonstrate our method is a universal optimization solution for existing Gaussian-based methods to improve novel view synthesis performance by about 1 dB PSNR for various tasks, including static object or scene reconstruction and dynamic 4D reconstruction.

## 2 RELATED WORK

**Volume Rendering.** Significant advancements have been achieved in novel-view synthesis, particularly since the introduction of NeRF (Barron et al., 2021; Mildenhall et al., 2021), which employs MLP to parameterize the geometry and view-dependent appearance with the help of implicitly defined radiance field. Moreover, the training and inference efficiency of NeRF has been enhanced using hash-grid (Müller et al., 2022) and explicitly defined samplers (Li et al., 2023). Built on top of the radiance field, NeuS (Wang et al., 2021), NeuS2 (Wang et al., 2023), and HF-NeuS (Wang et al., 2022) also perform more precise surface reconstruction against traditional MVS fusion such as MeshMVS (Shrestha et al., 2021). Given all the advantages of neural rendering, its efficiency is still not implausible. Recently, 3D Gaussian Splatting (3DGS) (Kerbl et al., 2023) has emerged, demonstrating impressive real-time NVS performance. Gaussian-based methods, typically represented by 3D Gaussian Splating (Kerbl et al., 2023), are recent advancements in 3D reconstruction, enabling high-quality and real-time rendering.

**Gaussian Splatting.** 3D Gaussian splitting Kerbl et al. (2023); Wu et al. (2024) rasterizes through $\alpha$-blending and depth-sorting to get Gaussians projected, thus achieving real-time rendering efficiency by avoiding the complex ray marching. Thanks to its real-time rendering speed and high-quality reconstruction performance, 3DGS has been improved and applied to numerous tasks, such as autonomous driving, reflective object reconstruction (Jiang et al., 2024), and 4D reconstruction (Wu et al., 2024). Subsequent works focus on improving Gaussian representation, such as techniques about low-pass filtering (Huang et al., 2024a) and structure grid representations (Lu et al., 2024). GaussianPro (Cheng et al., 2024) proposes a normal propagation method to bridge a gap from SfM initialization and mitigate densification limitations. Pixel-GS (Zhang et al., 2024b) proposes a gradient-based scaling densification strategy to avoid the generation of floater near the camera. However, these Gaussian-based explicit representation methods adopt a single-image optimization strategy (Mildenhall et al., 2021; Kerbl et al., 2023), leading to overfitting certain views and not robust to novel view synthesis, especially when challenging scenarios are encountered, *e.g.* dynamic, reflective, or few-shot reconstruction. Built on pre-trained networks, the extracted multi-view features can solve some of the mentioned difficulties. For example, MVSplat (Chen et al., 2024) builds a cost volume representation to store cross-view similarities for the estimation of depth. LatentSplat (Wewer et al., 2024) proposes a representation encoding uncertainty with latent 3D Gaussian features. PixelSplat (Charatan et al., 2024) predicts a dense probability distribution over 3D sampled Gaussian positions. In this paper, our proposed method provides a more general solution without relying on other pre-trained networks.

## 3 METHODOLOGY

Gaussian Splatting (Kerbl et al., 2023) is recently proposed for real-time novel-view synthesis and high-fidelity 3D geometric reconstruction. Instead of employing implicit representations such as density field in NeRF(Mildenhall et al., 2021) and SDF in NeuS (Wang et al., 2021), Gaussian Splatting leverages a set of anisotropic 3D Gaussians comprising their locations, colors, covariances, and opacities to parameterize a scene. Such an explicit representation dramatically improves the training and inference efficiency compared to previous methods like NeRF and NeuS. In the rendering process, Gaussian Splatting also adopts the point-based volume rendering technique (Kopanas et al., 2021; 2022a) following NeRF. As denoted in Fig. 2 (a), we diagram that NeRF cannot receive multi-view supervision in a training iteration due to its point-sampling strategy and implicit representation. The view-dependent radiance $\mathbf{C}(.)$ of each pixel $p$ in the image with the camera extrinsics $E$ and intrinsics $K$ is calculated by blending a set of 3D Gaussians along ray $r(p, E, K)$. While NeRF (Mildenhall et al., 2021) approximately blends with points assigned by a sampler (Li

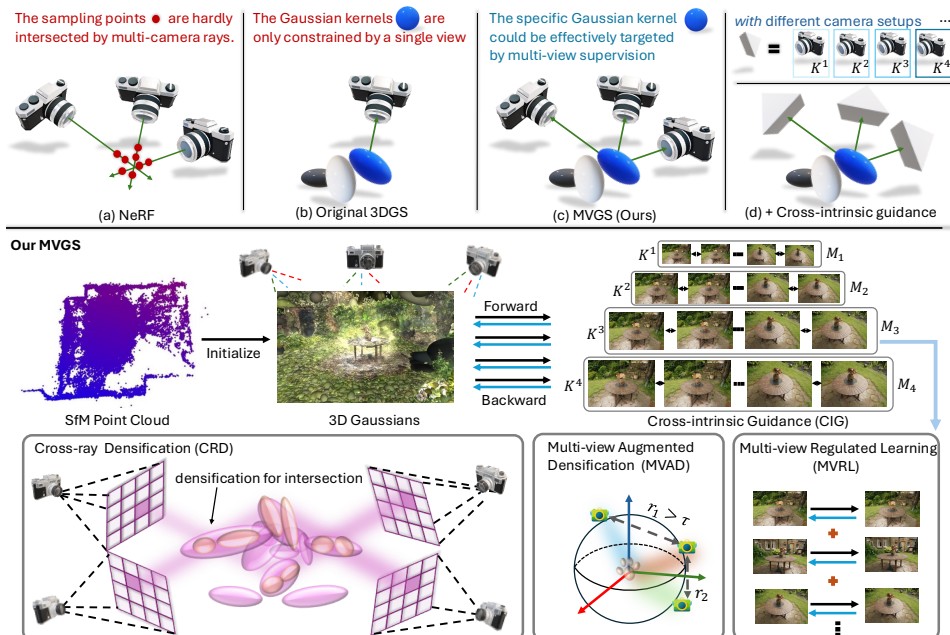

Figure 2: Illustration of the previous single-view training paradigm and our proposed *MVGS*, where (a) describes NeRF cannot be optimized in a multi-view training way. (b) points out the original 3DGS following the single-view training strategy of NeRF. (c) The proposed *MVGS* transforms the original training protocol followed by 3DGS and its variants. (d) The proposed cross-intrinsic guidance strategy enables multi-view training in a coarse-to-fine way. The bottom of this figure illustrates the pipeline of our proposed *MVGS*.

et al., 2023) in the radiance field, 3DGS precisely blends by rasterizing with $N$ parameterized kernels $\mathbf{G}(r) = \{g_i \mid i = 1, \ldots, N\}$ along ray $r(p, E, K)$. Assuming that the color $c_i \in \mathbb{R}^3$, the opacity $o_i \in \mathbb{R}$, and the covariance $\Sigma_i \in \mathbb{R}^{3 \times 3}$ describe the attributes of the $i$-th Gaussian $g_i$ respectively, the rendered pixel radiance $\mathbf{C}(r)$ is represented as

$$\mathbf{C}(r) = \mathbf{C}(\{g_i \mid i = 1, \ldots, N\}) = \sum_{i=1}^{N} c_i \alpha_i \prod_{j=1}^{i-1} (1 - \alpha_j) \ , \tag{1}$$

where the color $c_i$ is weighted by the transmittance $\alpha_i = o_i \exp\left(-\frac{1}{2}(x_i)^T \Sigma_i^{-1}(x_i)\right)$. Here $x_i$ denotes the distance between the position $\mu_i \in \mathbb{R}^3$ of Gaussian kernel and the query pixel $p$. $N$ represents the number of 3D Gaussians. Considering that 3DGS directly blends with the individually parameterized kernels $g_i = \{o_i, c_i, \Sigma_i\}$, which makes it possible to investigate optimization towards specific kernels across different views, we propose to improve the performance of 3DGS by a novel multi-view constraint in Sec. 3.1, stabilize rays by enriching different intrinsic setups in Sec. 3.2, and adapt the densification strategy in Sec. 3.3 and Sec. 3.4.

## 3.1 MULTI-VIEW REGULATED TRAINING

Given $T$ pairs of ground-truth images $\mathbf{I}$ and their corresponding camera extrinsics $\mathbf{E}$ and intrinsics $\mathbf{K}$, which are $\{(I_m, E_m, K_m) \mid m = 1, \ldots, T\}$, the goal of 3DGS is to reconstruct a 3D model described by the multi-view stereo data. As for the training strategy, 3DGS adheres to the convention of NeRF (Mildenhall et al., 2021) which optimizes the parametric model through single-view supervision per iteration. Regarding training, 3DGS is normally optimized by supervision from a single view of information per iteration, where the supervision in one iteration is randomly selected as $(I_i, E_i, K_i)$. Such that the loss function of the original 3DGS can be formulated accordingly as

$$\mathcal{L}(G, E_i, K_i) = \frac{1}{HW} \sum_{p=0}^{HW} (1 - \lambda) \mathcal{L}_1(I_i(p), \mathbf{C}(r(p, E_i, K_i))) + \lambda \mathcal{L}_{\text{D-SSIM}}(I_i(p), \mathbf{C}(r(p, E_i, K_i))),$$

$$\tag{2}$$

where $\mathcal{L}_1$ and $\mathcal{L}_{\text{D-SSIM}}$ denote the mean absolute error and D-SSIM loss (Kerbl et al., 2023), respectively. $G$ denotes partial $\mathbf{G}$ would be largely affected by gradients in single-view supervision mode. Practically, the hyperparameter $\lambda$ is used to control the proportion between these two loss terms.

Considering that implicit representations, *e.g.* NeRF, depend on pre-trained samplers to approximate the most confident blending points, multi-view supervision per iteration does not ensure improvement against single-view training, especially when the sampler is not well-trained as illustrated in Fig. 2 (a). The explicitly defined Gaussian kernels, on the other hand, do not depend on the sampler to get allocated as shown in Fig. 2 (b) which makes our proposed multi-view training strategy applicable as shown in Fig. 2 (c), where most of the blending kernels in $\mathbf{G}$ could be back-propagated with multi-view weighted gradients to overcome over-fitting problems towards certain perspectives.

Different from the original single-view iterative training, we propose a multi-view regulated training approach, optimizing 3D Gaussians in a multi-view supervision way. In particular, we sample $M$ pairs of supervised images and camera parameters at an iteration. Note that $M$ sets of matched images and camera parameters are sampled and different from each other. Therefore, our proposed multi-view regulated learning in a single iteration integrating in gradients can be represented as:

$$\frac{\partial \mathcal{L}}{\partial \{\mathbf{G}\}} = \frac{\partial \mathcal{L}(G_1, E_1, K_1)}{\partial G_1} + \frac{\partial \mathcal{L}(G_2, E_2, K_2)}{\partial G_2} + \cdots + \frac{\partial \mathcal{L}(G_M, E_M, K_M)}{\partial G_M}, \qquad (3)$$

where $\mathbf{G} = \{G_1, G_2, \ldots, G_M\}$ meaning that a portion of 3D Gaussians $G_M$ would be affected with large gradients for each view during multi-view training. The only difference with the original 3DGS loss is that our proposed method provides a multi-view constraint toward gradients for optimizing a set of 3D Gaussians $\mathbf{G}$. In this way, optimizing each Gaussian kernel $g_i$ would possibly get regulated by multi-view information so that over-fitting problems to certain views can be overcome. Moreover, the multi-view constraint enables 3D Gaussians to learn and deduce view-dependent information, like reflection as highlighted in the left part of Fig. 4, so our method can perform well in novel view synthesis for reflection scenes.

### 3.2 CROSS-INTRINSIC GUIDANCE

As shown in the bottom of Fig. 2, inspired by the benefits introduced by image pyramid (Adelson et al., 1984), we propose a coarse-to-fine training scheme with different camera setups, *i.e.* intrinsic parameters $K$, by simply supplementing more rasterization planes. Specifically, a 4-layer image pyramid with downsampling factors $\mathbf{S} = \{2^{k-1} \mid k = 4 \ldots 1\}$ could be constructed as shown in Fig. 2 (d). Empirically, the largest downsampling factor set as 8 is enough to accommodate sufficient training images for multi-view training and the smallest downsampling factor set as 1 means that the downsampling operation is not applied. For each layer, we have matched multi-view settings $M_s = \{M_1, M_2, M_3, M_4\}$. In particular, the larger downsampling factor enables more views accommodated to provide stronger multi-view constraints. In the initial three training stages, we run only a few thousand iterations per stage without completely training the model. Since target images are downsampled, the model cannot capture fine details during these early stages. Therefore, we treat the first three training stages as coarse training. During coarse training, incorporating more multi-view information imposes more powerful constraints on the entire 3D Gaussians. In this case, the rich multi-view information provides thorough supervision for the whole 3DGS and encourages fast fitting with coarse texture and structure. Once the coarse training is finished, fine training is started. Thanks to the previous coarse training stages providing a coarse architecture of 3DGS, the fine training stage only needs to refine and sculpt fine details for each 3D Gaussian. Especially, the coarse training stages provide a large number of multi-view constrain. It conveys the learned multi-view constraint to the next fine training. This scheme effectively enhances multi-view constraints and further improves the novel view synthesis performance.

### 3.3 CROSS-RAY DENSIFICATION

Due to the nature of volume rendering and the explicit representation of 3DGS, 3D Gaussians in some regions have a significant impact on distinct views when rendering. For instance, the central 3D Gaussians are crucial when rendering with cameras shooting the center in different poses. However, finding these regions is not trivial, especially in 3D space. As illustrated in Fig. 2, we propose a cross-ray densification strategy, starting from 2D space and then searching in 3D adaptively. Specifically,

we first calculate loss maps of multiple views and then locate the regions containing the largest average loss values with a sliding window with size $(h, w)$. Afterward, we cast rays from the vertices of these regions with four rays per window. Then, we calculate the intersection points across rays of different perspectives. Since we cast four rays per perspective, the intersection points can form several cuboids. These cuboids are the overlapped regions containing significant 3D Gaussians that play an important role when rendering for multiple views. Therefore, we densify more 3D Gaussians in these overlapped regions to facilitate the training of multi-view supervision. This strategy relies on the accurate searching of overlapped regions containing 3D Gaussians with high significance to several views. First, we choose the loss guidance, since it highlights the lowest-quality regions that should be improved for every view. Second, the ray casting technology allows us to locate the 3D regions containing a set of 3D Gaussians that contributes significantly to these views. Based on the accurate location, 3D Gaussians in these regions can be seen as pivotal for the joint optimization of multiple views. Note that we follow the densification mode of the original 3DGS to density one 3D Gaussian into two 3D Gaussians. In this way, we densify these 3D Gaussians to a certain amount to improve the reconstruction performance jointly for these views.

### 3.4 Multi-view Augmented Densification

To get fast convergence, avoid local minimum, and learn fine-grained Gaussian kernels while discrepancies between different views are significant, we propose a multi-view augmented densification strategy. Specifically, our strategy builds on the densification strategy of the original 3DGS with a predefined threshold $\beta$ used to determine which 3D Gaussians should be densified. As depicted in Fig. 2, we first identify whether the training views are strongly distinct. Instead of using the original camera translations directly, we normalize the camera translations of sampled views into a unit sphere. This approach makes our strategy adaptable to various scenes. Then, the relative translation distances $\{r_i \mid i = 1, 2, \ldots, n\}$ between each camera and another is computed, where the number of distances $n$ is $M^2 - M$ assuming that we have $M$ training views. In our multi-view augmented densification, we have a self-adaptive criterion $\hat{\beta}$ that can be formulated as

$$\hat{\beta} = \frac{\beta}{2} H\left(\frac{r_i}{\tau} - 1\right) + \beta\left(1 - H\left(\frac{r_i}{\tau} - 1\right)\right),\tag{4}$$

where $H(\cdot)$ is Heaviside function, returning 1 if the input is larger or equal 0. $\tau$ is a predefined hyperparameter. In this way, when the discrepancies between each view become large, the extent of 3D Gaussian densification is also enhanced. Consequently, our proposed multi-view augmented densification strategy allows 3D Gaussians to fit better for each view and capture finer details.

### 3.5 Implementation

In our experiments, we utilize novel view synthesis metrics like PSNR, SSIM, and LPIPS to evaluate the performance of models. For general object reconstruction, 3DGS (Kerbl et al., 2023) and Scaffold-GS (Lu et al., 2024) are selected as our baselines due to their state-of-the-art performance. For reflective object reconstruction, we choose 3DGS-DR (Ye et al., 2024) as our main baseline since it is the recent SOTA method to reconstruct glossy objects. As for 4D reconstruction, 4DGS (Wu et al., 2024) is selected as our baseline due to its fast rendering speed and high-quality 4D reconstruction performance. In large-scale scene reconstruction, Octree-GS (Ren et al., 2024) is adopted as our baseline since its level-of-detail structure is suitable for this kind of scene. In our proposed method, we set $M_s = \{48, 24, 12, 8\}$ and $\tau = 1$. As for the other setting, we follow the implementation setting of these baselines. Our method can be easily integrated into existing Gaussian-based methods without 200 lines of code.

## 4 Experiments

We conduct extensive experiments on various tasks that improve the performance of each baseline approach, ranging from static synthetic object-level scenes to indoor, outdoor, large-scale, and dynamic scenes. The validation results on each dataset prove that our method performs well especially in challenging cases, such as insufficient observations, texture-less area, view-dependent lighting effects, and fine-scale details.

Table 1: **Quantitative results of state-of-the-art 3D reconstruction methods on real-world datasets.** We report results on three commonly used datasets, including Mip-NeRF 360 (Barron et al., 2022), Tank&Temples (Knapitsch et al., 2017), and Deep Blending (Hedman et al., 2018). The best , second best , and third best results are denoted by red, orange, and yellow, respectively.

| Dataset | Mip-NeRF360 | | | Tanks&Temples | | | Deep Blending | | |
|---|---|---|---|---|---|---|---|---|---|
| Method & Metrics | PSNR↑ | SSIM↑ | LPIPS↓ | PSNR↑ | SSIM↑ | LPIPS↓ | PSNR↑ | SSIM↑ | LPIPS↓ |
| Instant-NGP (Müller et al., 2022) | 26.43 | 0.725 | 0.339 | 21.72 | 0.723 | 0.330 | 23.62 | 0.797 | 0.423 |
| Plenoxels (Fridovich-Keil et al., 2022) | 23.62 | 0.670 | 0.443 | 21.08 | 0.719 | 0.379 | 23.06 | 0.795 | 0.510 |
| Mip-NeRF 360 (Barron et al., 2022) | 29.23 | 0.844 | 0.207 | 22.22 | 0.759 | 0.257 | 29.40 | 0.901 | 0.245 |
| 2DGS(Huang et al., 2024b) | 28.98 | 0.867 | 0.185 | 23.43 | 0.845 | 0.181 | 29.70 | 0.902 | 0.250 |
| Fre-GS (Zhang et al., 2024a) | 27.85 | 0.826 | 0.209 | 23.96 | 0.841 | 0.183 | 29.93 | 0.904 | 0.240 |
| GES (Hamdi et al., 2024) | 28.69 | 0.857 | 0.206 | 23.35 | 0.836 | 0.198 | 29.68 | 0.901 | 0.252 |
| 3DGS (Kerbl et al., 2023) | 28.69 | 0.870 | 0.182 | 23.14 | 0.841 | 0.183 | 29.41 | 0.903 | 0.243 |
| Scaffold-GS (Lu et al., 2024) | 28.84 | 0.848 | 0.220 | 23.96 | 0.853 | 0.177 | 30.21 | 0.906 | 0.254 |
| 3DGS (**+Ours**) | 29.61 | 0.873 | 0.173 | 24.44 | 0.865 | 0.143 | 29.74 | 0.909 | 0.221 |
| Scaffold-GS(**+Ours**) | 29.82 | 0.877 | 0.171 | 25.54 | 0.902 | 0.093 | 30.37 | 0.915 | 0.153 |

Figure 3: **Qualitative comparisons of 3DGS (Kerbl et al., 2023), Scaffold-GS (Lu et al., 2024) and their improved version integrating our method across various datasets.** We use red close-up patches to highlight the visual differences for clearer visibility. We can observe that our proposed method can improve the original 3DGS and Scaffold-GS for extremely challenging scenes with strongly changed lighting effects, powerful reflection, and fine details.

**General Object Reconstruction.** To assess the performance of our proposed approach, we compare our improved version on 3DGS (Kerbl et al., 2023) and Scaffold-GS (Lu et al., 2024) baselines with their original methods. The quantitative results are shown in Table 1. As shown in Table 1, we conduct general object reconstruction experiments on three commonly used datasets, such as Mip-NeRF 360 (Barron et al., 2022), Tank&Temples (Knapitsch et al., 2017), and Deep Blending (Hedman et al., 2018). In Table 1, It can be observed that our method integrated into 3DGS and Scaffold-GS achieves SOTA results in terms of PSNR, SSIM, and LPIPS. In particular, Tank&Temples (Knapitsch et al., 2017) is a more challenging dataset than the others, containing more challenging scenes with the presence of texture-less regions, lighting changes, and reflections. As for qualitative comparisons, we present them in Fig. 3, showing the comparisons of 3DGS, Scaffold-GS, and their improvements by integrating our method. It can be observed that our method can improve the novel view synthesis performance quantitatively and qualitatively. In particular, previous methods are struggling to deal with scenes with strong reflection, fine details, and powerful lighting changes, leading to the phenomena of floaters, distortion, and over-smoothness. In contrast, our proposed multi-view regulated learning can impose multi-view constraints into the learning phase of 3D Gaussians so the trained model can interpolate novel views accurately. It indicates that previous methods integrated with our method can achieve better quantitative results and reconstruct more satisfying details.

**Reflective Object Reconstruction.** To demonstrate the universality of our proposed method, we conduct experiments for the reflective object reconstruction task. In particular, this task is more challenging than general object reconstruction because it contains objects with strong reflections and dramatic lighting effect changes. As depicted in Table 2, we compare several state-of-the-art reflective object reconstruction methods. Specifically, we conduct experiments on two commonly

Table 2: **Quantitative comparisons of state-of-the-art reflective object reconstruction methods.** We demonstrate our method can improve reconstruction performance for challenging reflection scenes. We report results on Shiny Blender and Glossy Synthetic datasets.

| Dataset | Shiny Blender | | | Glossy Synthetic | | |
|---|---|---|---|---|---|---|
| Method & Metrics | PSNR ↑ | SSIM↑ | LPIPS↓ | PSNR↑ | SSIM↑ | LPIPS↓ |
| Ref-NeRF (Verbin et al., 2022) | 33.12 | 0.961 | 0.079 | 27.49 | 0.927 | 0.100 |
| NPC (Kopanas et al., 2022b) | 27.48 | 0.921 | 0.145 | 21.96 | 0.841 | 0.181 |
| 3DGS (Kerbl et al., 2023) | 30.35 | 0.946 | 0.083 | 26.49 | 0.917 | 0.092 |
| GaussianShader (Jiang et al., 2024) | 31.96 | 0.957 | 0.067 | 27.53 | 0.921 | 0.086 |
| ENVIDR (Liang et al., 2023) | 33.46 | 0.967 | 0.045 | 29.56 | 0.952 | 0.059 |
| 3DGS-DR (Ye et al., 2024) | 34.08 | 0.971 | 0.052 | 30.13 | 0.953 | 0.058 |
| 3DGS-DR (**+Ours**) | 34.61 | 0.974 | 0.051 | 30.81 | 0.962 | 0.047 |

Table 3: **Quantitative results for 4D reconstruction on the D-NeRF (Pumarola et al., 2021) dataset.** We integrate our method into 4DGS and improve its 4D reconstruction performance. We also report the rendering speed (FPS) and storage size (MB) to demonstrate our method better. The rendering resolution is set to $800 \times 800$.

| Method | PSNR↑ | SSIM↑ | LPIPS↓ | FPS ↑ | Storage (MB)↓ |
|---|---|---|---|---|---|
| TiNeuVox-B (Fang et al., 2022) | 32.67 | 0.971 | 0.044 | 1.5 | 48 |
| KPlanes (Fridovich-Keil et al., 2023) | 31.61 | 0.974 | - | 0.97 | 418 |
| HexPlane-Slim (Cao & Johnson, 2023) | 31.04 | 0.973 | 0.044 | 2.5 | 38 |
| 3DGS (Kerbl et al., 2023) | 23.19 | 0.937 | 0.081 | 170 | 10 |
| FFDNeRF (Guo et al., 2023) | 32.68 | 0.973 | 0.041 | < 1 | 440 |
| MSTH (Wang et al., 2024) | 31.34 | 0.977 | 0.024 | - | - |
| 4DGS (Wu et al., 2024) | 34.05 | 0.978 | 0.023 | 82 | 18 |
| 4DGS (**+Ours**) | 35.11 | 0.980 | 0.021 | 102 | 12 |

used public datasets, like Shiny Blender (Verbin et al., 2022) and Glossy Synthetic dataset (Liu et al., 2023). In Table 2, it can be observed that our method integrated into 3DGS-DR achieves SOTA results compared with existing methods. In addition, we also present visual comparisons in the left part of Fig. 4 to assess our method qualitatively. We can find that 3DGS-DR cannot accurately recover lighting effects on glossy surfaces and fine details reflecting surrounding environments. In contrast, our method can reconstruct these details due to our proposed multi-view constraint. It is because our proposed multi-view regulated learning encourages the Gaussian-based explicit representation method following the constraint from multiple views to update and optimize the Gassuian attributes so that achieves better results. Moreover, it demonstrates our method can also be applied in reflection object reconstruction tasks and further indicates the universality of our proposed method.

**4D Reconstruction.** To further demonstrate the effectiveness of our proposed method, we conduct experiments for the task of 4D reconstruction. 4D reconstruction, known as dynamic scene reconstruction, is more challenging than 3D reconstruction since it contains the dimension of time, and the scenes are changed over time. In Table 3, we present detailed quantitative results on the D-NeRF (Pumarola et al., 2021) dataset for the evaluation of 4D reconstruction performance across state-of-the-art methods. It can be observed that our method integrated into 4DGS (Wu et al., 2024) achieves state-of-the-art results compared with existing state-of-the-art 4D reconstruction methods. In addition, we also report the rendering speed (FPS) and storage size (MB) metrics. We find that 4DGS integrated with our method achieves faster rendering speed with fewer 3D Gaussians. It indicates our method not only achieves better rendering performance but also faster rendering speed. The right part of Fig. 4 also demonstrates the effectiveness of our proposed method by reconstructing finer details. It is attributed to our proposed multi-view constraint method that constrains the optimization of 3D Gaussians with multi-view information, especially to dynamic scenes with temporal changed views.

**Large-scale Scene Reconstruction.** We additionally conduct experiments on a large-scale scene dataset, BungeeNeRF (Xiangli et al., 2022) to further prove the effectiveness of our method. As depicted in Table 4, we report results on three representative scenes. Note that our proposed method improves the recent SOTA Octree-GS for better novel view synthesis results. This improvement is due to the proposed multi-view training and densification strategies, constraining with multi-view supervision and generating more 3D Gaussians for faster convergence and finer details reconstruction.

Figure 4: **Qualitative results of 3DGS-DR (Ye et al., 2024), 4DGS (Wu et al., 2024) and their improved version by integrating our method across various challenging datasets.** It can be observed that 3DGS-DR and 4DGS integrated with our method can achieve better results for extremely challenging senses with strong reflection and dynamic changes.

Table 4: **Quantitative comparisons of state-of-the-art multi-scale scene reconstruction methods.** We demonstrate our method can also improve novel view synthesis performance for challenging multi-scale scenes. We report results on BungeeNeRF datasets (Xiangli et al., 2022).

| Scene | Chicago | | | Rome | | | Hollywood | | |
|---|---|---|---|---|---|---|---|---|---|
| Method & Metrics | PSNR↑ | SSIM↑ | LPIPS↓ | PSNR↑ | SSIM↑ | LPIPS↓ | PSNR↑ | SSIM↑ | LPIPS↓ |
| 3DGS | 28.17 | 0.930 | 0.084 | 27.54 | 0.916 | 0.100 | 26.24 | 0.869 | 0.133 |
| Mip-Splatting | 28.28 | 0.930 | 0.081 | 28.33 | 0.922 | 0.093 | 26.59 | 0.876 | 0.130 |
| Scaffold-GS | 28.55 | 0.929 | 0.080 | 28.24 | 0.924 | 0.087 | 26.36 | 0.866 | 0.157 |
| Octree-GS | 28.62 | 0.934 | 0.075 | 28.50 | 0.932 | 0.077 | 26.70 | 0.885 | 0.126 |
| Octree-GS (+Ours) | 28.82 | 0.936 | 0.069 | 28.79 | 0.933 | 0.073 | 26.73 | 0.887 | 0.122 |

This result also demonstrates that our method contains strong generalization to diverse scenes although they are not object-centered. The qualitative results can be found in the appendix.

## 4.1 ABLATION STUDY

To comprehensively demonstrate the effectiveness of our proposed method, we conduct ablation studies to evaluate the contributions of each component. As outlined in our method section, our proposed *MVGS* consists of four key components, such as multi-view regulated learning, cross-ray densification, multi-view augmented densification, and cross-intrinsic guidance. In our experiments, we find the appropriate multi-view training settings significantly improve rendering performance compared to existing Gaussian-based methods. This improvement is a distinguishing feature of our proposed method. As shown in Fig. 5, we compare existing state-of-the-art Gaussian methods with their counterparts enhanced by our proposed multi-view training. Fig. 5 investigates the relation between rendering improvements and the multi-view training settings. We observe that incorporating our multi-view training into existing methods leads to a substantial improvement in novel view synthesis quality. This enhancement is primarily attributed to our proposed multi-view regulated learning that constrains the optimization of the entire 3D Gaussians with multi-view information. However, when the number of multi-views increases to a certain number, the performance begins to

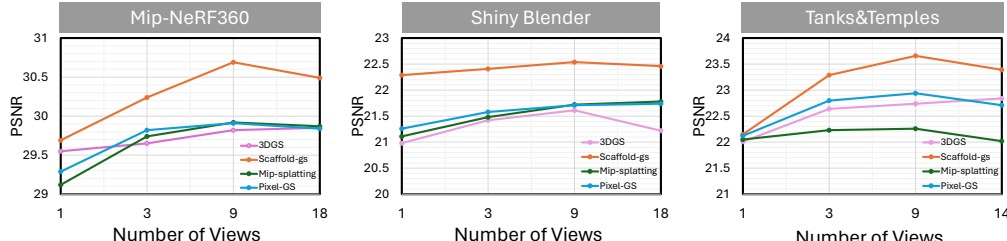

Figure 5: **Analysis of the multi-view training settings.** We improve four representative state-of-the-art Gaussian-based methods with the proposed multi-view regulated training. We report results on three representative datasets.

Table 5: **Detailed ablation studies across various Gaussian-based methods** We present the ablation studies on three state-of-the-art 3D reconstruction methods, 3DGS (Kerbl et al., 2023), Scaffold-GS (Lu et al., 2024), and Octree-GS (Ren et al., 2024). We report results on Mip-NeRF 360 dataset (Barron et al., 2022).

| Method | 3DGS | | | Scaffold-GS | | | Octree-GS | | |
|---|---|---|---|---|---|---|---|---|---|
| | PSNR | SSIM | LPIPS | PSNR | SSIM | LPIPS | PSNR | SSIM | LPIPS |
| Baseline | 28.70 | 0.905 | 0.204 | 29.34 | 0.914 | 0.191 | 29.70 | 0.911 | 0.183 |
| +Multi-view regulated learning | 29.76 | 0.919 | 0.174 | 30.46 | 0.923 | 0.167 | 30.34 | 0.913 | 0.171 |
| +Cross-ray densification | 29.86 | 0.920 | 0.170 | 30.59 | 0.925 | 0.165 | 30.42 | 0.921 | 0.168 |
| +Multi-view augmented densification | 30.14 | 0.926 | 0.153 | 30.86 | 0.926 | 0.159 | 30.51 | 0.922 | 0.158 |
| +Cross-intrinsic guidance (full) | 30.21 | 0.928 | 0.151 | 30.98 | 0.929 | 0.149 | 30.57 | 0.924 | 0.153 |

degrade. This occurs because an excessive number of multi-views leads to a large number of sampled views analogous to a region of views, encouraging 3D Gaussians to overfit in an area of the scene. Therefore, a moderate or scanty multi-view setting is more conducive to the optimization of 3DGS.

To further demonstrate the effectiveness of the proposed components, we conduct detailed ablation studies across various Gaussian-based 3D reconstruction methods as shown in Table 5. To be specific, we utilize three representative methods, including 3DGS (Kerbl et al., 2023), Scaffold-GS (Lu et al., 2024), and Octree-GS (Ren et al., 2024) as baselines and integrate our proposed method into them. As we can see in Table 5, the original performance of these baselines is inferior. When we incorporate the proposed multi-view regulated learning (MVRL) into baselines, the performance is improved by a huge margin. For example, the PSNR metric is improved over 1 dB for 3DGS and Scaffold-GS by our MVRL. This huge improvement is due to the proposed MVRL imposing multi-view constraints on the optimization of 3D Gaussians to enable 3D Gaussians robust for synthesizing more photorealistic results for novel views. In addition, we also propose two novel densification strategies, like cross-ray densification and multi-view augmented densification, to clone and split more 3D Gaussian primitives into appropriate regions for fitting better with the multi-view supervision. To fully leverage multi-view information, we propose cross-intrinsic guidance to train models with an image pyramid way for accommodating more views for multi-view training. With all of these proposed components, 3DGS can be improved over 1.5 dB. Scaffold-GS is also improved over 1.6 dB and Octree-GS gets improvement by over 0.8 dB. These results demonstrate the effectiveness of our proposed method and also indicate our method can improve existing methods to reach state-of-the-art performance.

## 5 CONCLUSION

In this work, we propose *MVGS*, a novel and universal method to improve the novel view synthesis performance for existing Gaussian-based methods. The core of *MVGS* lies in the proposed multi-view regulated learning, constraining the optimization of 3D Gaussians with multi-view information. We show that our method can be integrated into existing methods to achieve state-of-the-art rendering performance. We further demonstrate our proposed cross-intrinsic guidance scheme introducing powerful muti-view constraints for better results. We also prove the effectiveness of the proposed multi-view augmented densification and cross-ray densification in enhancing densification to facilitate the optimization of 3D Gaussians. Extensive experiments demonstrate the effectiveness of our method and indicate that our method achieves state-of-the-art novel view synthesis results.

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

## A  ADDITIONAL ABLATION STUDIES

We present additional ablation studies in Fig. 6. We leverage 3DGS as our baseline and integrate our proposed components progressively into it to demonstrate the effectiveness of the proposed methods. Specifically, the integration of the proposed multi-view regulated learning (MVRL) into 3DGS imposes the multi-view constraint for the optimization of the model to learn a more accurate multi-view structure. After that, we also progressively embed our proposed cross-ray densification (CRD) method into the baseline enforcing 3D Gaussians to split more primitives for better results. When the multi-view augmented densification (MVAD) is employed, the model has better capability to split more 3D Gaussians for arduous multi-view scenes. As we can see, the performance is improved by a huge margin. Finally, we integrate our proposed cross-intrinsic guidance (CIG) strategy, the model captures finer details in every scale training and obtains better results. These results demonstrate the effectiveness of our proposed components and indicate our method can be integrated into existing Gaussian-based methods for better novel view synthesis results.

## B  TRAINING WITH MORE ITERATIONS

For a faithful comparison, we also conduct an experiment to investigate the effect of training with more iterations. As illustrated in Fig. 7, we conduct experiments on three representative datasets, such as Mip-NeRF 360 (Barron et al., 2022), Shiny Blender (Verbin et al., 2022), and Tanks&Temples (Knapitsch et al., 2017) with 3DGS (Kerbl et al., 2023) as our baseline and its improved version by integrating with our proposed method. As seen in Fig. 7, the performance of 3DGS is obviously lower than ours. The original 3DGS method, despite being trained for more iterations, failed to reach the performance levels achieved by our proposed method. This indicates that mere increases in training duration do not compensate for the multi-view constraint absent in the original 3DGS. Our method not only speeds up the training convergence but also delivers massive performance improvement. These results indicate that the original 3DGS trained with more iterations cannot achieve performance improvement reaching like ours. It also demonstrates the significance of our proposed method.

## C  ADDITIONAL RESULTS ON 3D RECONSTRUCTION

In this section, we present additional experimental results for 3D reconstruction. To sufficiently demonstrate the effectiveness of our proposed method, we showcase per-scene quantitative results of the Mip-NeRF 360 dataset (Barron et al., 2022) in Table 6. As we can see in Table 6, 3DGS (Kerbl et al., 2023) and Scaffold-GS (Lu et al., 2024) integrated with our proposed method are better than their original performance. It demonstrates the effectiveness of our proposed method to improve 3D reconstruction results. We also present per-scene results of Tank&Temples (Knapitsch et al., 2017) and Deep Blending (Hedman et al., 2018) in Table 7. To be specific, we select representative scenes, including Truck and Train from Tank&Temples, and Playroom and Drjohnson from Deep Blending, respectively. It can be observed that our proposed method also demonstrates superior performance. In addition, we display additional visual comparisons of the task of 3D reconstruction in Fig. 8. We observe the original 3DGS and Scaffold-GS cannot recover details of the transparent surface or far objects. By integrating our proposed method, our proposed multi-view constraint encourages 3D Gaussians to capture finer details of multiple views and improve reconstruction quality.

## D  EXTRA COMPARISONS ON REFLECTIVE OBJECT RECONSTRUCTION

We also present additional experimental results and analysis for the task of reflective object reconstruction. As depicted in Table 8, we display per-scene quantitative results of Shiny Blender (Verbin et al., 2022) and Glossy Synthetic dataset (Liu et al., 2023). It is obvious that 3DGS-DR integrated with our proposed method achieves state-of-the-art results compared with other advanced methods. In addition, we also show visualization comparisons in Fig.9 and 10. Fig.9 displays additional visual comparisons of 3DGS-DR (Ye et al., 2024) with its improved version integrated with our proposed method. It demonstrates our proposed method can reconstruct better details on the glossy surface. Fig. 10 shows the reconstruction results of HDR and normal maps. HDR represents the environmental lighting effects of objects. The normal map can be seen as a geometry attribute that can indicate surface

reconstruction performance. As we can see in Fig. 10, our reconstructed HDR and normal maps are more approaching the references. It indicates that our reconstruction performance is better than the original 3DGS-DR. Thanks to our proposed multi-view constraint method leveraging multi-view information to constraint the optimization of the whole 3D Gaussians so that obtain better results.

## E  ADDITIONAL QUANTITATIVE AND QUALITATIVE RESULTS ON 4D RECONSTRUCTION

To further substantiate the effectiveness of our approach, we provide additional results for 4D reconstruction. As illustrated in Table 9, we offer per-scene quantitative results of D-NeRF (Pumarola et al., 2021), which serve to comprehensively analyze the performance improvements brought by our proposed method. The data clearly indicates that, when integrated into 4DGS (Wu et al., 2024), our method achieves state-of-the-art results in 4D reconstruction, marking a significant leap in reconstructing fidelity and accuracy. In addition to the quantitative analysis, we also present visual comparisons in Fig. 11 to further evaluate the qualitative performance of our method. As depicted in the visual results, 4DGS on its own struggles to reconstruct fine details, often failing to capture subtle textures and intricate structural elements. In contrast, our approach, when integrated into 4DGS, yields a substantial improvement, enabling the reconstruction of much finer details with clearer and more accurate texture representation. These visual results, alongside the quantitative improvements, demonstrate that our method not only enhances the clarity and sharpness of the reconstructed scenes but also significantly reduces artifacts and inaccuracies, leading to a more realistic and lifelike representation of 4D dynamic scenes.

The combination of qualitative and quantitative evidence strongly supports the superiority of our method over existing approaches. Our method consistently leads to performance improvements across a wide range of tasks, proving its versatility and robustness in enhancing 4D reconstruction. Furthermore, the ability of our approach to adapt to different scenes and capture intricate details in dynamic reconstructions showcases its potential for a broad array of applications in areas such as motion capture, virtual reality, and high-fidelity simulations. The results solidify the contribution of our method to progress the state-of-the-art in 4D reconstruction.

## F  VISUALIZATION OF MULTI-SCALE SCENE RECONSTRUCTION

We also provide the visualization results of multi-scale scene reconstruction on the BungeeNeRF dataset (Xiangli et al., 2022) in Fig.12. Upon analysis, we observe that the original Octree-GS(Ren et al., 2024) struggles to reconstruct intricate details and tends to produce noticeable artifacts, which hinders its overall reconstruction quality. In contrast, the enhanced version of Octree-GS, empowered by our proposed method, successfully captures and renders finer texture details, closely resembling the ground truth. This improvement underscores the robustness and precision of our approach, as it significantly reduces artifacts while enhancing visual quality across complex scenes.

These experiments offer comprehensive evidence that not only highlights the effectiveness of our method but also demonstrates its ability to generalize well to a wide variety of scenes. This includes both simple and complex environments, reinforcing the applicability of our approach across different reconstruction tasks. Moreover, the results indicate that our method consistently improves rendering quality, particularly for novel view synthesis. Its versatility extends across multiple applications such as general object reconstruction, reflective object reconstruction, 4D dynamic scene reconstruction, and multi-scale scene reconstruction. These findings emphasize the broad potential of our approach in advancing the state-of-the-art Gauissian-based methods for novel view synthesis.

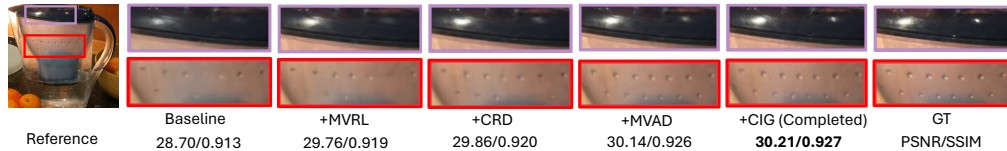

| Reference | Baseline
28.70/0.913 | +MVRL
29.76/0.919 | +CRD
29.86/0.920 | +MVAD
30.14/0.926 | +CIG (Completed)
**30.21/0.927** | GT
PSNR/SSIM |

Figure 6: **Visualization comparisons of the ablation of the proposed components.** We employ 3DGS as our baseline and improve it by gradually integrating our proposed components into it. It can be observed our method gradually improves the novel view synthesis performance of the baseline.

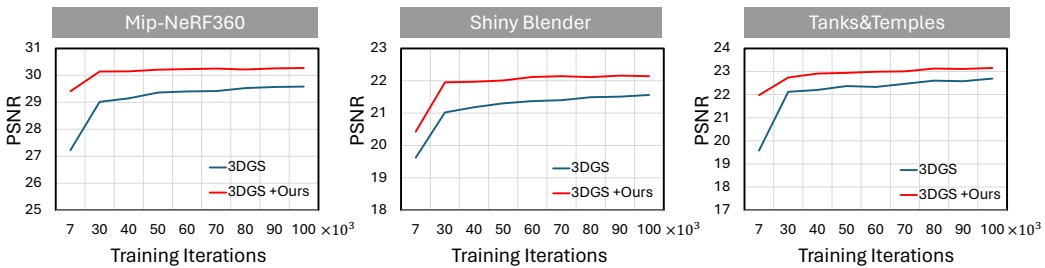

Figure 7: **Study on the effect of additional training iterations.** We leverage state-of-the-art 3DGS (Kerbl et al., 2023) as our baseline and conduct experiments on three representative datasets, such as Mip-NeRF 360 (Barron et al., 2022), Shiny Blender (Verbin et al., 2022), and Tanks&Temples (Knapitsch et al., 2017).

Table 6: **Detailed quantitative results of state-of-the-art 3D reconstruction methods on Mip-NeRF 360 dataset (Barron et al., 2022)**. The best , second best , and third best results are denoted by red, orange, and yellow, respectively.

| Metrics
3D Scenes | PSNR ↑ | SSIM↑
Stump | LPIPS↓ | PSNR↑ | SSIM↑
Room | LPIPS↓ | PSNR↑ | SSIM↑
Counter | LPIPS↓ | PSNR↑ | SSIM↑
bonsai | LPIPS↓ |
|---|---|---|---|---|---|---|---|---|---|---|---|---|
| Mip-NeRF 360 (Barron et al., 2022) | 26.40 | 0.744 | 0.261 | 31.63 | 0.913 | 0.211 | 29.55 | 0.894 | 0.204 | 33.46 | 0.941 | 0.176 |
| 3DGS (Kerbl et al., 2023) | 26.55 | 0.775 | 0.210 | 30.63 | 0.914 | 0.220 | 28.70 | 0.905 | 0.204 | 31.98 | 0.938 | 0.205 |
| Scaffold-GS (Lu et al., 2024) | 26.27 | 0.784 | 0.284 | 31.93 | 0.925 | 0.202 | 29.34 | 0.914 | 0.191 | 32.70 | 0.946 | 0.185 |
| 3DGS (+Ours) | 26.39 | 0.760 | 0.243 | 32.84 | 0.932 | 0.184 | 30.21 | 0.928 | 0.151 | 33.05 | 0.949 | 0.167 |
| Scaffold-GS(+Ours) | 26.74 | 0.775 | 0.232 | 33.08 | 0.935 | 0.174 | 30.98 | 0.929 | 0.149 | 33.69 | 0.953 | 0.163 |

| 3D Scenes | | Bicycle | | | Garden | | | Kitchen | | | | |
|---|---|---|---|---|---|---|---|---|---|---|---|---|
| Mip-NeRF 360 (Barron et al., 2022) | 24.37 | 0.685 | 0.301 | 26.98 | 0.813 | 0.170 | 32.23 | 0.920 | 0.127 | | | |
| 3DGS (Kerbl et al., 2023) | 25.25 | 0.771 | 0.205 | 27.41 | 0.868 | 0.103 | 30.32 | 0.922 | 0.129 | | | |
| Scaffold-GS (Lu et al., 2024) | 24.50 | 0.705 | 0.306 | 27.17 | 0.842 | 0.146 | 31.30 | 0.928 | 0.126 | | | |
| 3DGS (+Ours) | 25.08 | 0.752 | 0.226 | 27.23 | 0.856 | 0.123 | 32.57 | 0.934 | 0.113 | | | |
| Scaffold-GS(+Ours) | 25.23 | 0.760 | 0.226 | 27.48 | 0.855 | 0.124 | 31.96 | 0.933 | 0.114 | | | |

Table 7: **Detailed quantitative comparisons of state-of-the-art 3D reconstruction methods on Tank&Temples (Knapitsch et al., 2017) and Deep Blending (Hedman et al., 2018).** We choose two challenging scenes, Truck and Tran from the Tank&Temples dataset for evaluation. As for Deep Blending, we select two representative scenes, Playroom and Drjohnson for assessment.

| Dataset | | | Tanks&Temples | | | | | Deep Blending | | | | |
|---|---|---|---|---|---|---|---|---|---|---|---|---|
| 3D Scenes
Method | PSNR ↑ | SSIM↑
Truck | LPIPS↓ | PSNR↑ | SSIM↑
Train | LPIPS↓ | PSNR↑ | SSIM↑
Playroom | LPIPS↓ | PSNR↑ | SSIM↑
Drjohnson | LPIPS↓ |
| 3DGS (Kerbl et al., 2023) | 25.18 | 0.879 | 0.148 | 21.09 | 0.802 | 0.218 | 30.04 | 0.906 | 0.241 | 28.77 | 0.899 | 0.244 |
| Mip-NeRF 360 (Barron et al., 2022) | 24.91 | 0.857 | 0.159 | 19.52 | 0.660 | 0.354 | 29.66 | 0.900 | 0.252 | 29.14 | 0.901 | 0.237 |
| Scaffold-GS (Lu et al., 2024) | 25.77 | 0.883 | 0.147 | 22.15 | 0.822 | 0.206 | 30.62 | 0.904 | 0.258 | 29.80 | 0.907 | 0.250 |
| 3DGS (+Ours) | 26.14 | 0.893 | 0.125 | 22.74 | 0.838 | 0.162 | 30.33 | 0.927 | 0.201 | 29.16 | 0.892 | 0.241 |
| Scaffold-GS(+Ours) | 27.19 | 0.926 | 0.071 | 23.88 | 0.878 | 0.116 | 30.84 | 0.925 | 0.152 | 29.91 | 0.905 | 0.154 |

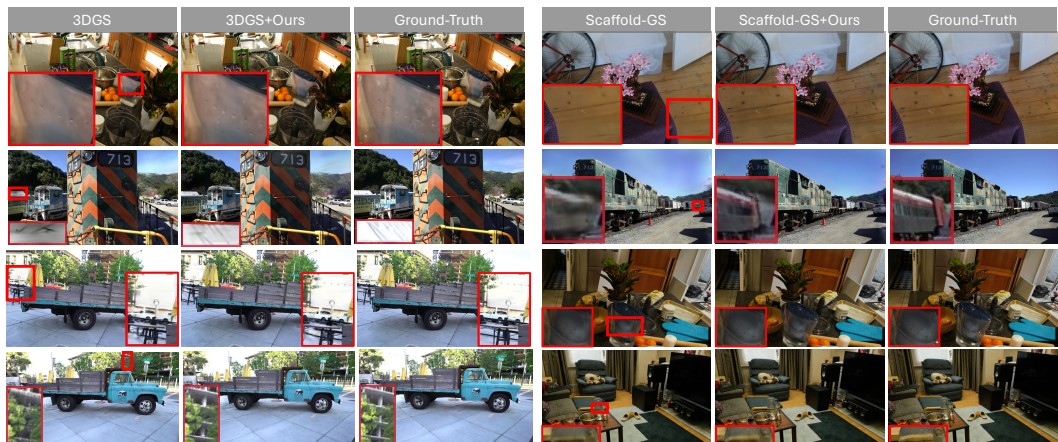

Figure 8: **Additional qualitative comparisons of general object reconstruction. We compare 3DGS (Kerbl et al., 2023) and Scaffold-GS (Lu et al., 2024) with their improved version by integrating our method across various datasets.** We employ red close-up patches to highlight the visual differences for better differentiation. It can be observed that our proposed method can improve the original 3DGS and Scaffold-GS for challenging scenes.

Table 8: **Detailed quantitative results of state-of-the-art reflective object reconstruction methods.** We report PSNR, SSIM, and LPIPS metrics on each scene from Shiny Blender (Verbin et al., 2022) and Glossy Synthetic (Liu et al., 2023).

| Datasets | | Shiny Blender | | | | | | Glossy Synthetic | | | | | |
|---|---|---|---|---|---|---|---|---|---|---|---|---|---|
| | | ball | car | coffee | helmet | teapot | toaster | bell | cat | luyu | potion | tbell | teapot |
| PSNR ↑ | Ref-NeRF (Verbin et al., 2022) | 33.16 | 30.44 | 33.99 | 29.94 | 45.12 | 26.12 | 30.02 | 29.76 | 25.42 | 30.11 | 26.91 | 22.77 |
| | NPC (Kopanas et al., 2022b) | 23.76 | 24.19 | 30.39 | 25.59 | 41.22 | 19.76 | 22.41 | 25.35 | 23.68 | 23.09 | 19.03 | 18.21 |
| | 3DGS (Kerbl et al., 2023) | 27.65 | 27.26 | 32.3 | 28.22 | 45.71 | 20.99 | 25.11 | 31.36 | 26.97 | 30.16 | 23.88 | 21.51 |
| | GShader (Jiang et al., 2024) | 30.99 | 27.96 | 32.39 | 28.32 | 45.86 | 26.28 | 28.07 | 31.81 | 27.18 | 30.09 | 24.48 | 23.58 |
| | ENVIDR (Liang et al., 2023) | 41.02 | 27.81 | 30.57 | 32.71 | 42.62 | 26.03 | 30.88 | 31.04 | 28.03 | 32.11 | 28.64 | 26.77 |
| | 3DGS-DR (Ye et al., 2024) | 33.66 | 30.39 | 34.65 | 31.69 | 47.12 | 27.02 | 31.65 | 33.86 | 28.71 | 32.29 | 28.94 | 25.36 |
| | 3DGS-DR (+Ours) | 34.51 | 30.83 | 34.81 | 32.24 | 47.93 | 27.36 | 33.20 | 33.93 | 29.31 | 32.90 | 29.31 | 26.91 |
| SSIM ↑ | Ref-NeRF (Verbin et al., 2022) | 0.971 | 0.950 | 0.972 | 0.954 | 0.995 | 0.921 | 0.941 | 0.944 | 0.901 | 0.933 | 0.947 | 0.897 |
| | NPC (Kopanas et al., 2022b) | 0.908 | 0.898 | 0.955 | 0.938 | 0.994 | 0.835 | 0.892 | 0.921 | 0.854 | 0.877 | 0.742 | 0.762 |
| | 3DGS (Kerbl et al., 2023) | 0.937 | 0.931 | 0.972 | 0.951 | 0.996 | 0.894 | 0.908 | 0.959 | 0.916 | 0.938 | 0.900 | 0.881 |
| | GShader (Jiang et al., 2024) | 0.966 | 0.932 | 0.971 | 0.951 | 0.996 | 0.929 | 0.919 | 0.961 | 0.914 | 0.936 | 0.898 | 0.901 |
| | ENVIDR (Liang et al., 2023) | 0.997 | 0.943 | 0.962 | 0.987 | 0.995 | 0.922 | 0.954 | 0.965 | 0.931 | 0.960 | 0.947 | 0.957 |
| | 3DGS-DR (Ye et al., 2024) | 0.979 | 0.962 | 0.976 | 0.971 | 0.997 | 0.943 | 0.962 | 0.976 | 0.936 | 0.957 | 0.952 | 0.936 |
| | 3DGS-DR (+Ours) | 0.983 | 0.965 | 0.976 | 0.974 | 0.998 | 0.949 | 0.974 | 0.979 | 0.947 | 0.963 | 0.965 | 0.942 |
| LPIPS ↓ | Ref-NeRF (Verbin et al., 2022) | 0.166 | 0.050 | 0.082 | 0.086 | 0.012 | 0.083 | 0.102 | 0.104 | 0.098 | 0.084 | 0.114 | 0.098 |
| | NPC (Kopanas et al., 2022b) | 0.237 | 0.120 | 0.119 | 0.156 | 0.013 | 0.226 | 0.203 | 0.121 | 0.101 | 0.174 | 0.243 | 0.246 |
| | 3DGS (Kerbl et al., 2023) | 0.162 | 0.047 | 0.079 | 0.081 | 0.008 | 0.125 | 0.104 | 0.062 | 0.064 | 0.093 | 0.125 | 0.102 |
| | GShader (Jiang et al., 2024) | 0.121 | 0.044 | 0.078 | 0.074 | 0.007 | 0.079 | 0.098 | 0.056 | 0.064 | 0.088 | 0.122 | 0.091 |
| | ENVIDR (Liang et al., 2023) | 0.020 | 0.046 | 0.083 | 0.036 | 0.009 | 0.081 | 0.054 | 0.049 | 0.059 | 0.072 | 0.069 | 0.041 |
| | 3DGS-DR (Ye et al., 2024) | 0.098 | 0.033 | 0.076 | 0.049 | 0.005 | 0.081 | 0.046 | 0.040 | 0.053 | 0.075 | 0.067 | 0.067 |
| | 3DGS-DR (+Ours) | 0.089 | 0.030 | 0.074 | 0.042 | 0.004 | 0.067 | 0.031 | 0.035 | 0.044 | 0.062 | 0.048 | 0.060 |

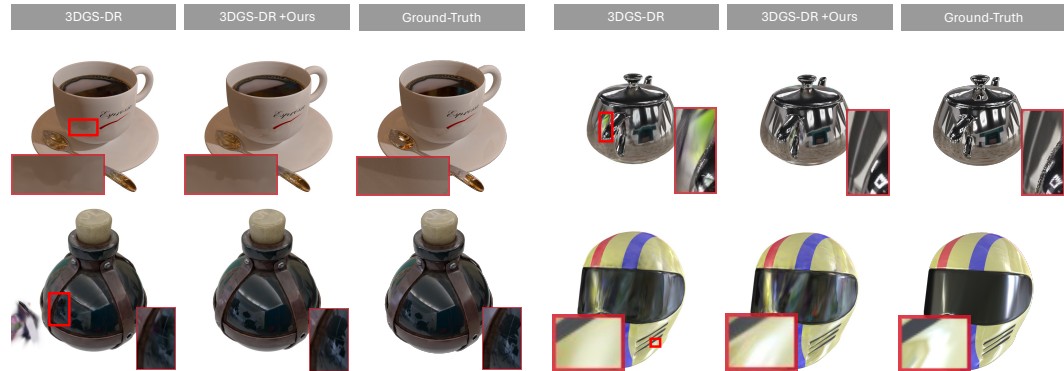

Figure 9: **Additional qualitative results of 3DGS-DR (Ye et al., 2024) and our enhanced version across diverse reflective object datasets.** We can find that 3DGS-DR enhanced by our proposed method can be more robust for challenging scenes with reflection effects to obtain better reflective reconstruction performance.

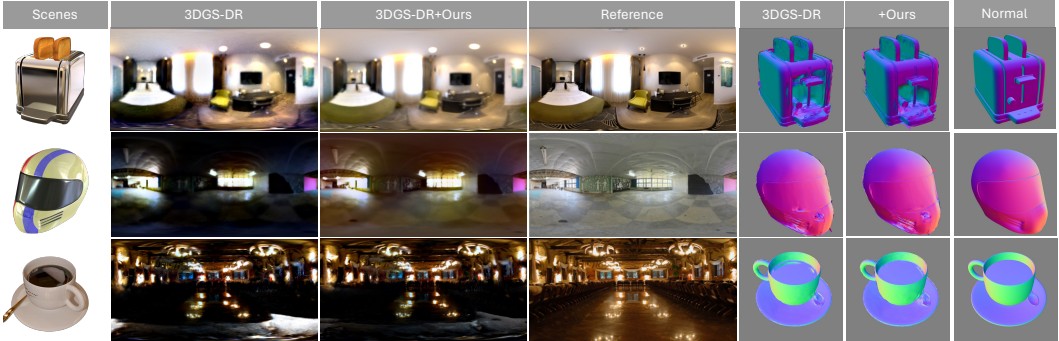

Figure 10: **Qualitative comparisons of HDR reconstruction and normal reconstruction by 3DGS-DR (Ye et al., 2024) and our proposed method.** The better performance of HDR and normal reconstruction means better reflective object reconstruction performance.

Table 9: **Per-scene quantitative results for 4D reconstruction on the D-NeRF (Pumarola et al., 2021) dataset.** We integrate our method into 4DGS and improve its 4D reconstruction performance.

| Method | Bouncing Balls | | | Hellwarrior | | | Hook | | | Jumpingjacks | | |
|---|---|---|---|---|---|---|---|---|---|---|---|---|
| | PSNR | SSIM | LPIPS | PSNR | SSIM | LPIPS | PSNR | SSIM | LPIPS | PSNR | SSIM | LPIPS |
| 3DGS | 23.20 | 0.959 | 0.060 | 24.53 | 0.933 | 0.058 | 21.71 | 0.887 | 0.103 | 23.20 | 0.959 | 0.060 |
| K-Planes | 40.05 | 0.993 | 0.032 | 24.58 | 0.952 | 0.082 | 28.12 | 0.948 | 0.066 | 31.11 | 0.970 | 0.046 |
| HexPlane | 39.86 | 0.991 | 0.032 | 24.55 | 0.944 | 0.073 | 28.63 | 0.957 | 0.050 | 31.31 | 0.972 | 0.039 |
| TiNeuVox | 40.23 | 0.992 | 0.041 | 27.10 | 0.963 | 0.076 | 28.63 | 0.943 | 0.063 | 33.49 | 0.977 | 0.040 |
| 4DGS | 40.62 | 0.994 | 0.015 | 28.71 | 0.973 | 0.036 | 32.73 | 0.976 | 0.027 | 35.42 | 0.985 | 0.012 |
| 4DGS + (**Ours**) | 41.60 | 0.995 | 0.011 | 29.29 | 0.976 | 0.029 | 33.67 | 0.979 | 0.021 | 37.69 | 0.990 | 0.011 |

| Method | Lego | | | Mutant | | | Standup | | | Trex | | |
|---|---|---|---|---|---|---|---|---|---|---|---|---|
| | PSNR | SSIM | LPIPS | PSNR | SSIM | LPIPS | PSNR | SSIM | LPIPS | PSNR | SSIM | LPIPS |
| 3DGS | 23.06 | 0.929 | 0.064 | 20.64 | 0.929 | 0.082 | 21.91 | 0.930 | 0.078 | 21.93 | 0.953 | 0.048 |
| K-Planes | 25.49 | 0.948 | 0.033 | 32.50 | 0.971 | 0.036 | 33.10 | 0.979 | 0.031 | 30.43 | 0.973 | 0.034 |
| HexPlane | 25.10 | 0.938 | 0.043 | 33.67 | 0.9802 | 0.026 | 34.40 | 0.983 | 0.020 | 30.67 | 0.974 | 0.027 |
| TiNeuVox | 24.65 | 0.906 | 0.064 | 30.87 | 0.960 | 0.047 | 34.61 | 0.979 | 0.032 | 31.25 | 0.966 | 0.047 |
| 4DGS | 25.03 | 0.937 | 0.038 | 37.59 | 0.988 | 0.016 | 38.11 | 0.989 | 0.007 | 34.23 | 0.985 | 0.013 |
| 4DGS (**+Ours**) | 24.70 | 0.932 | 0.057 | 38.82 | 0.991 | 0.012 | 40.81 | 0.993 | 0.008 | 34.26 | 0.985 | 0.019 |

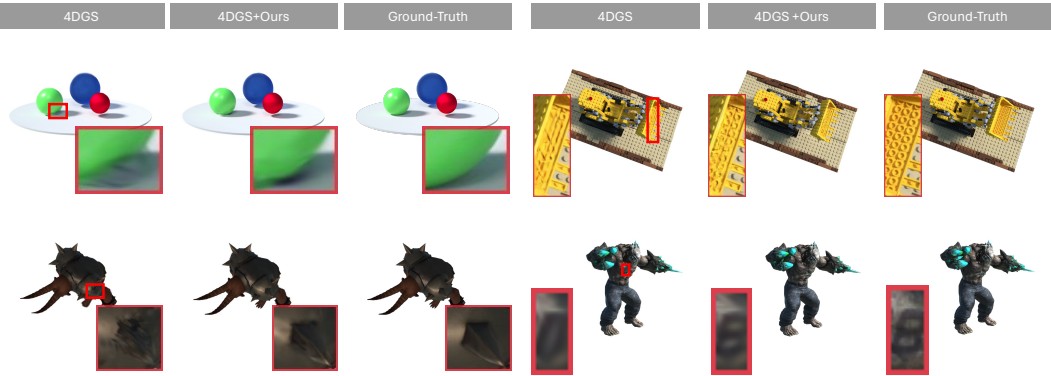

Figure 11: **Additional visualization comparisons of 4DGS (Wu et al., 2024) and its improved version integrating with our method for 4D reconstruction.** It can be found that our proposed method can enhance 4DGS to reconstruct finer dynamic details and obtain better performance.

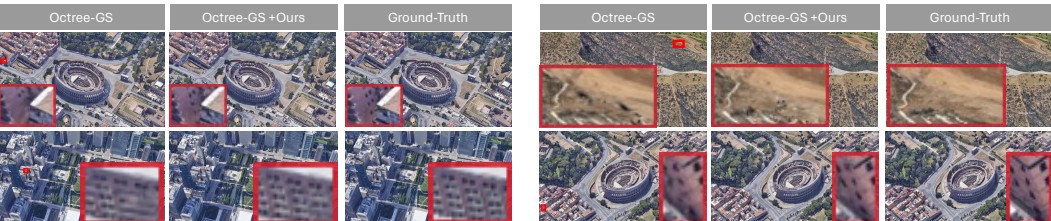

Figure 12: **Qualitative comparisons of multi-scale scene reconstruction on BungeeNeRF dataset (Xiangli et al., 2022). We compare Octree-GS (Ren et al., 2024) and its improved version by integrating our method.** We utilize red close-up patches to stand out the visual differences for clear comparisons. We can find that our proposed method can improve the original Octree-GS for challenging multi-scale scenes.

