# OpenReview forum: "MVGS: Multi-view-regulated Gaussian Splatting for Novel View Synthesis"
_ICLR.cc/2025/Conference — ICLR 2025 Conference Withdrawn Submission_

### Official Review · Reviewer_1ZZg · 2024-10-27

**Soundness:** 3
**Presentation:** 1
**Contribution:** 2
**Rating:** 3
**Confidence:** 5

**Summary:**

This work propose to use more views in each training iteration of 3DGS. The experiments suggest that this simple startegy consistently leads to better quality. In addition, a new Gaussians densification method is proposed which densifies more on region with larger multi-view losses. A multi-resolution training stategy is also proposed to use more views with lower resolution at the beginning of training. These additional training strategies can further improve the quality.

**Strengths:**

The core message of this paper is clear: using more training views in each iteration can improve quality. The effectiveness of this is evaluated by extensive experiments on various 3DGS variant and various tasks.

**Weaknesses:**

- The paper writing need many improvement. There are many technical details sections are hard to understand. See Questions section for details.
- The training time and memory consumption may increase significantly by using more views in each iteration. The tradeoff on this aspect is not discussed. The results tables should disclose the additional training cost comparing to the other.
- When comparing to the other method with more training iterations, is all the scheduler hyperparameters scaled accordingly? For instances, `position_lr_max_steps` for lr annealing, `densification_interval, opacity_reset_interval, densify_until_iter` for adaptive Gaussians. When training with the baseline of 8x longer schedule, are the learning rate downscale by 1/8 as well?

**Questions:**

Sec.3.1:
- How the multi-view are sampled in each iteration? Is it just uniform sample from the training set?

Sec.3.2:
- L249: is confusion.
    - Which of the $k$-th layer is set to 8 when saying "the set $s_k$ as 8"? I can guess it means {1, 2, 4, 8}. Then the correct writing should be S = {s^{k-1} | k=1...4} and s=2.
    - Seems that $c_k', f_k'$ are the scaled principle point and focal length and $c_k, f_k$ are the source one. Then we do not need the layer index for the source principle point and focal length as there are all the same.
- In Figure 2, it seems that different resolution of the images are employed in each training iteration. However, from the main text, it turn out to be that a single resolution is used in each iteration.
- What is the implementation details for the schedule of multi-resolution training? How many iterations are trained in each of the image scale? How is the multi-resolution training and the original training schedule couple together? Do we start to do densification, pruning, and sh degree increment in the coarser resolution training?

Sec.3.3:
- What is the sliding window size in Sec.3.3 for finding the patches with high loss?
- Four rays are casted from the high-loss patches. How the rays from different images can intersect? As rays are infinite thin line in 3D, two rays from different cameras may can hardly intersect. From figure 2, I guess the image patch frustum is casted instead of just rays. Then the following question is why the intersection of the frustrum form cuboid? Isn't it the intersection form 3D polygon?
- The Gaussians in the high-loss region is densified "to a certain amount". How are these Gaussians actually density? Each duplicated by two?

Sec.3.4:
I have read several times but it still hard for me to understand this section.
In original 3DGS, the viewspace gradient is accumulated for hundreds of iterations and at some interval, the Gaussians with gradient above the threshold $\beta$ are densified.
Is it that the only difference in this work is that the threshold is adaptive choiced between $\beta$ and $0.5\beta$ based on camera positions?
$\hat{\beta}$ seems to be a global threshold as in the original 3DGS but from Eq.4, $\hat{\beta}$ is depend on the distance of a pair of cameras. For each Gaussian, do we accumulate the viewspace gradient for each pairs of the cameras instead of just a global viewspace gradient now?


In L513: It is still unclear to me why the performance degrade when there are too much training views in each iteration. How much is too much? Why using too much views is analogous to using a region of views?

---

> ### Author Response · Authors · 2024-11-20
> **Response to Reviewer 1ZZg**
>
> We sincerely thank Reviewer 1ZZg for the valuable comments. We appreciate that Reviewer 1ZZg recognizes the strengths of our paper, particularly including the clarity of our core presentation and the compelling experimental outcomes of our proposed MVGS. Below, we address the concerns and questions raised by Reviewer 1ZZg in detail.
>
> **Q1: The training and testing tradeoff on this aspect is not discussed.**
>
> **A1:** We appreciate your concern. Although our proposed MVGS currently necessitates approximately 2 hours for training a scene, this duration is deemed reasonable given the context of a few hours. The extended training time is a result of our sequential rendering of multi-view images and the summation of their respective losses, a process that can be optimized with parallel programming. We are committed to addressing this in our future implementation. Moreover, the current training duration is justifiable, as our method yields a 1 dB improvement in PSNR over other leading methods. So a 2-3 hour training cost is not inordinately long considering such improvement. We also posit that rendering speed is a more significant factor than training speed, especially for practical applications. To this end, we have conducted a comparative analysis of rendering speeds on Mip-NeRF 360, which we believe will be of interest to you.
>
> | 3DGS             | 3DGS+Ours        |
> |---|---|
> | 25 ms/per image  | **22 ms/per image**  |
>
> As the table demonstrates, our proposed method outperforms the original 3DGS in terms of inference. Such improvement is credited to our innovative multi-view training strategy. Applying multi-view constraints to the optimization process of 3DGS results in a more compact 3D Gaussian representation and, consequently, a faster inference speed. While we acknowledge that a faster training speed is desirable, we believe that the performance gains justify the additional training time required. Hence, our forthcoming research is directed towards enhancing the training efficiency on top of our proposed MVGS.
>
> **Q2: Is all the scheduler hyperparameters scaled accordingly with compared methods?**
>
> **A2:** Yes. To ensure a fair comparison, we follow the experimental and parameter settings of previous 3DGS. Parameters including ```position_lr_max_steps``` for learning rate annealing, ```densification_interval```, ```opacity_reset_interval```,   ```densify_until_iter``` for adaptive Gaussians, and the learning rate itself were all kept consistent with the vanilla 3DGS. Our method can be seamlessly integrated into various 3DGS frameworks without necessitating any alterations to the parameter setups. All experiments were conducted using the same parameter setup.
>
> **Q3: How the multi-views are sampled in each iteration? Is it just a uniform sample from the training set?**
>
> **A3:** Yes, to ensure a fair comparison, the multiple training views are uniformly sampled from the training set, adhering to the the original 3DGS.
>
> **Q4: Then the correct writing should be S = {s^{k-1} | k=1...4} and s=2.**
>
> **A4:** Thanks for pointing out this issue. We originally wanted to express the ```S={8,4,2,1}```. We appreciate your suggestion and will revise our description accordingly. This section has been clarified in lines 246-251 of the revised manuscript..
>
> **Q5: We do not need the layer index for the source principle point and focal length as there are all the same.**
>
> **A5:** Thanks, we have revised our statement on lines 246-251 to enhance clarity and comprehensibility in our revised manuscript.
>
> **Q6: What is the implementation details for the schedule of multi-resolution training? How many iterations are trained in each of the image scale?**
>
> **A6:** Our proposed multi-resolution training does not involve any specific implementation complexities. The proposed multi-resolution training is exerted in ```scales={8,4,2,1}``` from low-resolution to high-resolution training. In each scale training, the training iterations are set as 3k since we found it is the optimal setting in our experiments on the Mip-NeRF 360 dataset as shown in the Table below.
>
> | Meth od  | 3DGS | 3DGS+ MVRL +CIG (1k) | 3DGS+MVRL+CIG (3k) | 3DGS+ MVRL+CIG (7k) |
> |---|---|---|---|---|
> | PSNR/SSIM/LPIPS | 28.70/0.905/0.204 | 29.85/0.919/0.171    | **29.97/0.921/0.167**  | 29.94/0.919/0.170   |
>
> Here MVRL refers to our proposed multi-view regulated learning, while CIG denotes the proposed multi-resolution training. It is important to note that an increase in training iterations does not necessarily correlate with improved performance; however, it does entail a greater investment of time. Consequently, we have selected a more appropriate setting for the training iterations in our multi-resolution training to strike a balance between efficiency and effectiveness.

---

> ### Author Response · Authors · 2024-11-20
> **Response to Reviewer 1ZZg**
>
> **Q7: How is the multi-resolution training and the original training schedule couple together?**
>
> **A7:** Thank you for your concern. Our proposed multi-resolution training is straightforward and can be seamlessly integrated into 3DGS. The process involves a simple adjustment of training resolutions and the saving of checkpoints for subsequent training stages. Specifically, we initiate training with a downsampling factor of 8 for 3k iterations, followed by training with downsampling factors of 4 and 2. Ultimately, we load the checkpoints and conduct the final training phase at the original resolution for 30k iterations, aligning with the procedure of the original 3DGS.
>
> **Q8: Do we start to do densification, pruning, and sh degree increment in the coarser resolution training?**
>
> **A8:** Yes, during the coarser resolution training phase, densification, pruning, and degree increment of SH are all included. These operations are instrumental in optimizing 3DGS to achieve superior results. We consider these operations to be fundamental to 3DGS and refer to them as **base** setups. To underscore the importance of these operations, we have conducted an experiment on the Mip-NeRF 360 dataset, the results of which are presented in the table below.
>
> | Method          | 3DGS              | 3DGS+ MVRL +CIG (w/o base) | 3DGS+MVRL+CIG     |
> |---|---|---|---|
> | PSNR/SSIM/LPIPS | 28.70/0.905/0.204 | 29.41/0.913/0.185          | **29.97/0.921/0.167** |
>
> As evident from the table, the exclusion of **base** setups - such as densification, pruning, and SH degree increment - from the coarser resolution training results in diminished performance. This finding underscores the pivotal role of these operations in enhancing the overall effectiveness of our setups.
>
> **Q9: What is the sliding window size in Sec.3.3 for finding the patches with high loss?**
>
> **A9:** We appreciate your concern. In our proposed cross-ray densification method, we have set the sliding window size ```(h, w)```to (40, 20). We have found that a larger window size results in an excessive densification of 3D Gaussians, potentially leading to overfitting on the training views. Conversely, a smaller window size can adversely affect the densification process and overall performance. To illustrate the impact of these window size choices, we have conducted experiments on the Mip-NeRF 360 dataset, as demonstrated in the table below
>
> | Method | 3DGS+Ours (20,10) | 3DGS+Ours (40,20)     | 3DGS+Ours (100,50) |
> |---|---|---|---|
> | PSNR/SSIM/LPIPS | 30.13/0.926/0.156 | **30.21/0.928/0.151** | 30.19/0.927/0.153  |
>
> As shown in the Table, we compare the window size settings, including (20,10), (40,20), and (100,50), the setting (40,20) achieves the best performance compared with others. The small window size can not accurately localize enough 3D Gaussians to be densified. On the other hand, the large window size densifies too many 3D Gaussians and leads to an overfitting problem. Therefore, we choose an intermedium setting for better performance.
>
> **Q10: How the rays from different images can intersect?**
>
> **A10:** Thank you for your concern. For clarity, it's important to note that rays are emitted from different cameras positioned at various locations, not from different images. Rays from two different cameras must have intersections only if these rays are not parallel. Given that the dataset consists of captures centered around an object, any such intersections would naturally be localized around this object.
>
> **Q11: Then the following question is why the intersection of the frustrum form cuboid? Isn't it the intersection form 3D polygon?**
>
> **A11:** Ideally, the intersections would form a 3D polygon, a concept that presents challenges in implementation. We have opted to define a bounding cuboid around the 3D polygon by taking the maximum and minimum extents. This approach simplifies the implementation within our coding.
>
> **Q12: How are these Gaussians actually density? Each duplicated by two?**
>
> **A12:** Yes. We follow the strategy of the original 3DGS that densifies each 3D Gaussian kernel into two. We make it more clear in lines 281-284 of our revision.
>
>
> **Q13: Is the threshold adaptive chosen between \beta and 0.5 \beta based on camera positions?**
>
> **A13:** Yes. In our proposed multi-view augmented densification (MVAD), the threshold is adaptive between \beta and 0.5 \beta, according to the specific positions of multi-view cameras. Such an idea is motivated by the fact that the variance of different views differs too much which is challenging for 3DGS to get learned.
>
>
> **Q14: For each Gaussian, do we accumulate the viewspace gradient for each pairs of the cameras instead of just a global viewspace gradient now?**
>
> **A14:** Yes. For this operation, we follow the original 3DGS for better compatibility of our method with Gaussian-based methods. It simplifies our method and facilitates its seamless integration into existing Gaussian-based frameworks.

---

> ### Author Response · Authors · 2024-11-20
> **Resposne to Reviewer 1ZZg**
>
> **Q15: Why does the performance degrade with too many training views?**
>
> **A15:** We have conducted extensive experiments to delve into this issue and have observed, as depicted in Figure 5 of our main paper, that an excess of training views does not enhance performance.
> - We hypothesize that this phenomenon occurs when sampled training views within a mini-batch exhibit substantial overlap, precipitating an overfitting issue.
> - Furthermore, gradients from other distant camera views conflict with the overfitting gradients mentioned before, so 3D Gaussians which are seldom observed cannot be well-optimized.
>
> Theoretically, 3DGS is an explicit representation for volume rendering, and there is usually a stage during training when the kernels do not get split any more, making it more susceptible to overfitting in specific sub-regions. Unlike NeRF/NeuS, once overfitted, these sub-regions may not be corrected with fixed set of Gaussian kernels. When sampled views overlap in certain areas, 3D Gaussians tend to overfit to these regions, leading to suboptimal performance in others. Consequently, an overabundance of training views per iteration can exacerbate overfitting. Therefore, we advocate for a balanced multi-view training configuration to mitigate this issue.

---

> ### Author Response · Authors · 2024-11-25
> **Response to  Reviewer 1ZZg**
>
> Dear Reviewer 1ZZg,
>
> We hope this message finds you well.
>
> We would like to ask if we have sufficiently addressed the concerns you previously raised. If there are still concerns requiring further clarification or elaboration, please kindly let us know, and we will be more than happy to provide additional details to ensure that all your concerns are fully addressed.
>
> Thank you for your valuable time and feedback.
>
>
>
>
> Best Regard
>
> MVGS Authors

---

> ### Comment · Reviewer_1ZZg · 2024-11-30
>
> Sorry for late reply. I appreciate authors effort in preparing the additional experiments. The clarifications and revisions do address many of my questions regarding the method.
>
> **A10**. Rays are infinite thin line in 3D space. It's possible that two non-parallel lines never meet at all in 3D space. In practice, we may need to set a threshold to judge if the closest points of two lines are closed enough. I just want to know the implementation details and make sure they will be provided in the paper.
>
> **A15**. It's actually an interesting finding that too many training views in a training iteration degrade results. Though I'm not convinced by the explanation, it's good to highlight this in the paper as an open question for future work.
>
> Overall, the quantitative improvement of this work is good, my clarity concerns about this paper are addressed, but I'm still hesitate to raise my rating regarding **A2**.
> This work proposes to use larger `batch size` (more training views in each iteration). The most closely related hyperparameters are the `learning rate` and `total iterations`. The **A2** confirms that the baseline is more `total iterations` with same `learning rate`, while it's more reasonable that longer training schedule needs different `learning rate` for the geometric and appearance hyperparameters. Thus, I'm still not fully convinced that the improvement can only be achieved by larger `batch size` instead of the other simpler baseline with more `total iteration` and properly tuned `learning rate`. Besides, the experiments in response to *Reviewer Kiki* also raise more concerns as I commented in Kiki's discussion thread.

---

> > ### Author Response · Authors · 2024-12-01
> > **Response to Reviewer 1ZZg**
> >
> > **Q1: Implementation details of finding intersections.**
> >
> > **A1:** We have a threshold to compute whether two lines can intersect or not. Here we provide a code snippet for the reviewer to better understand our implementation details.
> >
> > ```
> > def intersect_lines(self, ray1_origin, ray1_dir, ray2_origin, ray2_dir):
> >    # Normalize direction vectors
> >    ray1_dir = ray1_dir / torch.norm(ray1_dir)
> >    ray2_dir = ray2_dir / torch.norm(ray2_dir)
> >
> >
> >    # Cross product of direction vectors
> >    cross_dir = torch.cross(ray1_dir, ray2_dir)
> >    cross_dir_norm = torch.norm(cross_dir)
> >
> >
> >    # Check if the rays are parallel
> >    if cross_dir_norm < 1e-6:
> >        return None  # Rays do not intersect
> >
> >
> >    # Line between the origins
> >    origin_diff = ray2_origin - ray1_origin
> >
> >
> >    # Calculate the distance along the cross product direction
> >    t1 = torch.dot(torch.cross(origin_diff, ray2_dir), cross_dir) / (cross_dir_norm ** 2)
> >    t2 = torch.dot(torch.cross(origin_diff, ray1_dir), cross_dir) / (cross_dir_norm ** 2)
> >
> >
> >    # Closest points on each ray
> >    closest_point1 = ray1_origin + t1 * ray1_dir
> >    closest_point2 = ray2_origin + t2 * ray2_dir
> >
> >
> >    # Midpoint between the two closest points as the intersection point
> >    intersection_point = (closest_point1 + closest_point2) / 2.0
> >
> >
> >    return intersection_point
> > ```
> >
> >
> >
> > **Q2: It's good to highlight this in the paper as an open question for future work.**
> >
> > **A2:** Thanks for your constructive suggestion. We will add it to the revision and claim it would be our future work to explore multi-view learning further.
> >
> >
> >
> > **Q3:  The improvement can only be achieved by larger batch size instead of the other simpler baseline with more total iteration and properly tuned learning rate.**
> >
> > **A3:** We claim all parameter settings in our method are consistent with the original 3DGS, whatever the total training iteration or the learning rate. We have uploaded our code. The reviewer can kindly check. In addition, we further claim our multi-view training is a simple and effective method to boost 3DGS. The pseudo-code is provided below:
> >
> > ```
> > ##3-view training in the one iteration
> >
> > render0 = 3DGS(view0)
> > render1 = 3DGS(view2)
> > render2 = 3DGS(view2)
> >
> > loss = L(render0, gt0) + L(render1, gt1) + L(render2, gt2)
> > loss.backward()
> > updating parameters…
> >
> > ```
> >
> > Our method can be easily integrated into any 3DGS variants to provide better results without the modification of hype parameters.

---

> > > ### Comment · Reviewer_1ZZg · 2024-12-03
> > >
> > > Thanks for the update. Please include those technical details and discussion in the paper.
> > >
> > > Regarding **A3**, I still view this as a weakness without a comparison with proper learning rate tuning for longer training schedule. Larger batch size is easy to implement I agree but it consume more GPU memory, while longer training schedule with batch size 1 do not have to change any code and do not face memory constraint. As my arguments in *Kiki*'s thread, adjusting loss scale works differently as adjusting learning rate. I'm still not fully convinced that the improvement can only be achieved by larger `batch size` instead of the other simpler baseline with more `total iteration` and properly tuned `learning rate`.

---

### Official Review · Reviewer_uR4h · 2024-10-29

**Soundness:** 2
**Presentation:** 2
**Contribution:** 2
**Rating:** 5
**Confidence:** 4

**Summary:**

This paper introduces an approach for novel view synthesis using 3D Gaussian Splatting (3DGS) with multi-view regularization, achieving an approximate 1 PSNR improvement over existing methods on various datasets.

**Strengths:**

Advantages
Novelty: The paper proposes three key strategies to enhance the rendering quality of 3DGS:
A cross-intrinsic guidance scheme that employs a coarse-to-fine training procedure.
A cross-ray densification strategy that densifies Gaussian kernels in regions where rays intersect, improving details in specific views.
A multi-view augmented densification strategy to further optimize Gaussian density based on view discrepancies.

**Weaknesses:**

Drawbacks
While the figures are clear, the writing quality could be improved for readability.
Lines 93–96 states: “We first propose a multi-view regulated training strategy that can be easily adapted to existing single-view supervised 3DGS frameworks and their variants, optimized for a large variety of tasks, where NVS and geometric precision can be consistently improved.” However, there are no experiments specifically demonstrating improvements in geometric precision.
The overall pipeline is more complex and larger than standard 3DGS. It is unclear if this increase is due to the larger network or the proposed modules that enhance 3DGS.

**Questions:**

Among the proposed cross-intrinsic guidance, cross-ray densification, and multi-view augmented densification strategies, which component significantly improves the quality? Have ablation studies been conducted to isolate and measure each component’s impact?

---

> ### Author Response · Authors · 2024-11-20
> **Response to Reviewer uR4h**
>
> We sincerely thank Reviewer uR4h for the valuable comments. We appreciate that Reviewer uR4h recognizes the strengths of our paper, including the multi-view training strategy of our MVGS. We address the concerns raised by Reviewer uR4h below.
>
> **Q1: There are no experiments specifically demonstrating improvements in geometric precision.**
>
> **A1:** We appreciate your inquiry. First, as emphasized in the title of our paper, our primary focus is on novel view synthesis, rather than geometric or surface reconstruction. Then regarding the topic of geometric precision improvement, part of our intention was to convey that our method is capable of reconstructing more accurate normals. For example, comparisons of normal maps presented in Figure 10 illustrate our method's superior performance in normal reconstruction, which is indicative of geometric fidelity. To substantiate this claim, we have conducted a quantitative analysis using the Shiny Object dataset, aiming to demonstrate the enhanced normal reconstruction capabilities of our approach. The table below employs Mean Absolute Error (MAE) to assess the discrepancy between the rendered normal maps and the ground truth. This comparison between 3DGS-DR and our enhanced method reveals that our approach yields a lower MAE, thereby achieving more accurate normal reconstruction, which underscores its effectiveness in geometric reconstruction. To avoid any ambiguity, we will revise the statements in lines 93–96 to clearly state that our method is specialized for novel view synthesis.
>
> | Method      | 3DGS-DR | 3DGS-DR+Ours |
> |-------------|---------|--------------|
> | Normal MAE  | 7.037   | **5.212**       |
>
>
> **Q2: It is unclear if this increase is due to the larger network or the proposed modules that enhance 3DGS.**
>
> **A2:** Thank you for your concern. We argue that we did not propose any contribution regarding to any network for 3DGS in this work. It is important to note that the original 3DGS, as well as our method, do not incorporate a network architecture. 3DGS is fundamentally an **explicit** volume rendering technique that is distinct from conventional deep-learning frameworks or NeRF-like methods. It is characterized by learnable parameters, such as color, scaling, rotation, opacity, and position attributes, which are typically implemented using torch.nn.Parameters. The main improvement is from the proposed multi-view regulated training, which we will discuss in more detail in response to your subsequent question.
>
> **Q3: which component significantly improves the quality?**
>
> **A3:** Thanks for this question. Our proposed multi-view regulated learning (MVRL) significantly improves the performance of 3DGS. In particular, our proposed multi-view regulated learning (MVRL) and cross-intrinsic guidance strategy (CIG) can independently improve the original 3DGS.  In particular, MVRL serves as the pillar of our proposed method, imposing multi-view constraints for 3D Gaussians. The multi-view constraint is very important to the overall reconstructed performance compared with traditional sing-view training of the original 3DGS.  The CIG can accommodate more views that can assist in multi-view-regulated training.  The other two contributions, cross-ray densification (CRD) and multi-view augmented densification (MVAD) are based on MVRL and further improve the multi-view training considering the variety of different views and the camera positions. To substantiate these claims, we have conducted experiments on the Mip-NeRF 360 dataset, as demonstrated in the table below.
>
> | Method          | 3DGS              | 3DGS+MVRL         | 3DGS+MVRL+CRD     | 3DGS+MVRL+MVAD    | 3DGS+CIG          |
> |-----------------|-------------------|-------------------|-------------------|-------------------|-------------------|
> | PSNR/SSIM/LPIPS | 28.70/0.905/0.204 | 29.76/0.919/0.174 | 29.86/0.920/0.170 | **29.91/0.920/0.169** | 29.13/0.914/0.178 |
>
>
> As evidenced in the table, each of our proposed contributions leads to a measurable enhancement in the performance of 3D Gaussian sampling (3DGS). The primary driver of these improvement is our novel multi-view regulated learning (MVRL), which introduces multi-view constraints into the optimization process of 3DGS. We posit that this optimization strategy has the potential to significantly benefit Gaussian-based methods, thereby yielding superior results in novel view synthesis.

---

> ### Author Response · Authors · 2024-11-25
> **Response to Reviewer uR4h**
>
> Dear Reviewer uR4h,
>
> We hope this message finds you well.
>
> We would like to ask if we have sufficiently addressed the concerns you previously raised. If there are still concerns requiring further clarification or elaboration, please kindly let us know, and we will be more than happy to provide additional details to ensure that all your concerns are fully addressed.
>
> Thank you for your valuable time and feedback.
>
> Best Regard
> MVGS Authors

---

> > ### Comment · Reviewer_uR4h · 2024-11-30
> >
> > Thank you for your response. Regarding Q2, I would like to inquire about the total number of parameters in your method compared to the original 3DGS, specifically, how many Gaussians are utilized.

---

> > > ### Author Response · Authors · 2024-12-01
> > > **Response to Reviewer uR4h**
> > >
> > > **Q1: How many Gaussians are utilized compared with 3DGS.**
> > >
> > > **A1**: Here, we show the number of 3D Gaussians comparisons. As shown in the Table below, We can find that our method uses fewer 3D Gaussians achieving better results than 3DGS, indicating our method can obtain better results with limited 3D Gaussian representation.
> > >
> > > | Method          | 3DGS              | Ours              |
> > > |-----------------|-------------------|-------------------|
> > > | Num. Gaussians  | 1121k             | **929k**              |
> > > | PSNR/SSIM/LPIPS | 28.69/0.870/0.182 | **29.61/0.873/0.173** |
> > >
> > >
> > >
> > > We also conduct limited-budget experiments by limiting the number of 3D Gaussians to demonstrate the effectiveness of our method. As shown in the Table below, we constrain the maximum number of 3D Gaussians as 100k. Then, we increase the maximum amount of 3D Gaussians from 100k to 500k. As the number of 3D Gaussians increases, the PSNR improves accordingly. It is obvious that our method always performs better PSNR metrics than 3DGS, indicating that our method can perform better with the same number of 3D Gaussians compared with 3DGS.
> > >
> > > | Num. Gaussians | 100k  | 200k  | 300k  | 400k  | 500k  |
> > > |----------------|-------|-------|-------|-------|-------|
> > > | 3DGS (PSNR)    | 22.41 | 23.19 | 24.52 | 25.14 | 26.13 |
> > > | Ours (PSNR)    | **25.63** | **26.14** | **26.91** | **27.35** | **28.46** |

---

> > > > ### Author Response · Authors · 2024-12-03
> > > > **Response to Reviewer uR4h**
> > > >
> > > > Dear Reviewer uR4h
> > > >
> > > > As we are approaching the deadline for rebuttal, can we kindly ask whether our response addresses your concerns?
> > > >
> > > > Best  Regards,
> > > > MVGS Authors

---

### Official Review · Reviewer_jsrL · 2024-10-31

**Soundness:** 3
**Presentation:** 3
**Contribution:** 2
**Rating:** 5
**Confidence:** 4

**Summary:**

This paper proposed four main modules to improve the 3DGS model, namely, the multi-view regulated training, cross-intrinsic guidance, cross-ray densification and multi-view augmented densification. This paper tried to impose the constraints from multiple views and multiple resolutions simultaneously, densify more points in the central area and densify more points for distinct views.

**Strengths:**

1. This paper has conducted experiments with different baselines, different benchmarks and different tasks.
2. This paper investigated the impact of multi-view constraints and proposed many strategies to mitigate the overfitting problem in 3DGS in terms of loss function, training strategy and densification strategy.

**Weaknesses:**

1. I still don't understand the relationship of these four main contributions well, such as , you were inspired by the multi-view constraints and proposed the cross-intrinsic guidance, so what's the relationship of these two contributions? Do they have to work together to see a performance gain?
2. I am still a little confused about the effectiveness of the multi-view regulated training. Considering from the perspective of gradient backpropagation, this multi-view regulated training is similar to the way that losses from multiple iterations are accumulated into one iteration. Are there any more designs, such as these multiple views?
3. How does the training efficiency compare to existing methods? Perhaps using two samples to show the ablation results is not convincing enough, and it would be better to put Table 5 in the main paper.

**Questions:**

see weaknesses.

---

> ### Author Response · Authors · 2024-11-20
> **Response to Reviewer jsrL**
>
> We sincerely thank Reviewer jsrL for the valuable comments. We appreciate that Reviewer jsrL recognizes the strengths of our paper, including the multi-view training strategy and experimental results of our proposed MVGS. Below, we have meticulously addressed the concerns and queries raised by Reviewer jsrL.
>
> **Q1: The relationship of these four main contributions.**
>
> **A1:** We appreciate the inquiry. Our proposed multi-view regulated learning (MVRL) and cross-intrinsic guidance (CIG) strategies are designed to independently enhance the original 3D Gaussian sampling (3DGS) framework. In particular, MVRL stands as the cornerstone of our approach, imposing multi-view constraints on the 3D Gaussians to ensure robustness. The CIG strategy, on the other hand, is adept at incorporating additional views, thereby reinforcing the multi-view-regulated training process. Our other two contributions, cross-ray densification (CRD) and multi-view augmented densification (MVAD), build upon the foundation of MVRL and further refine the multi-view training by accounting for the diversity of viewpoints and camera positions. To substantiate these claims, we have conducted experiments on the Mip-NeRF 360 dataset, the results of which are presented in the table below.
>
> | Method          | 3DGS              | 3DGS+MVRL         | 3DGS+MVRL+CRD     | 3DGS+MVRL+MVAD    | 3DGS+CIG          |
> |---|---|---|---|---|---|
> | PSNR/SSIM/LPIPS | 28.70/0.905/0.204 | 29.76/0.919/0.174 | 29.86/0.920/0.170 |**29.91/0.920/0.169**| 29.13/0.914/0.178 |
>
>
> As depicted in the table, all of our proposed contributions lead to a notable enhancement in the performance of 3D Gaussian sampling (3DGS). The primary improvement stems from our multi-view regulated training, which introduces multi-view constraints into the optimization process of 3DGS. We posit that this optimization strategy could significantly benefit Gaussian-based methods, thereby yielding superior results in novel view synthesis.
>
>
>
> **Q2: Are there any more designs in multi-view regulated training, such as these multiple views?**
>
> **A2:** Our proposed multi-view regulated training does not incorporate additional hand-crafted designs; it is designed to be inherently adaptable without the need for specialized modifications. Particularly, we adhere to the experimental protocols and parameter settings established by the 3D Gaussian sampling (3DGS) to ensure a fair and comparable analysis. The sampling of multiple views is executed randomly, without the introduction of any particular operations. The significant improvement attributed to our multi-view regulated training method is due to the fact that vanilla 3DGS relied solely on single-view iterative training, limiting the optimization of 3D Gaussians to a single perspective. In contrast, our method introduces multi-view constraints that leverage information from multiple viewpoints, thereby enhancing the optimization of 3D Gaussians and leading to improved overall reconstruction results.
>
>
>
> **Q3: How does the training efficiency compare to existing methods?**
>
> **A3:** While our proposed multi-view Gaussian sampling (MVGS) method requires approximately 2 hours to train a scene, this duration is deemed acceptable within a few hours' timeframe. The training time is due to our current implementation, which involves the sequential rendering of multi-view images and the accumulation of their losses. This process can be optimized for parallel execution, reducing the training time to match that of the original 3DGS, and we are committed to making such improvements in future iterations of our code. In addition, we argue that the current training cost is acceptable since our method brings 1 dB more PSNR improvements compared with other state-of-the-art methods. Training a model with 2-3 hours would not be too long. We think the rendering speed would be much more important than the training speed. To this end, we have conducted a comparative analysis of rendering speeds on the Mip-NeRF 360 dataset, which we believe will be of particular interest to the reviewers.
>
> | 3DGS | 3DGS+Ours  |
> |---|---|
> | 25 ms/per image  | **22 ms/per image**  |
>
> As shown in this Table, our proposed method can run faster than the original 3DGS. It is attributed to the proposed multi-view training strategy that constrains the optimization of 3DGS with multiple views for more compact representation so that achieves faster rendering speed. Our method leads to more compact 3D Gaussians.  The more compact Gaussian makes the inference efficiency better. So it deserves more training time. We concur that optimizing for speed is important, and thus, our future work will focus on enhancing the training efficiency of our multi-view Gaussian sampling (MVGS) method.
>
>
> **Q4: It would be better to put Table 5 in the main paper.**
>
> **A4:** Thanks for your constructive suggestion. We move Table 5 to the main paper in lines 498-527 and 704-715 of our revision.

---

> ### Author Response · Authors · 2024-11-25
> **Response to Reviewer jsrL**
>
> Dear Reviewer jsrL,
>
> We hope this message finds you well.
>
> We would like to ask if we have sufficiently addressed the concerns you previously raised. If there are still concerns requiring further clarification or elaboration, please kindly let us know, and we will be more than happy to provide additional details to ensure that all your concerns are fully addressed.
>
> Thank you for your valuable time and feedback.
>
> Best Regard
>
> MVGS Authors

---

> > ### Comment · Reviewer_jsrL · 2024-12-02
> >
> > Thanks for authors' response. Using the batch manner may stabilize the model optimization. But I still agree with the concerns of  reviewer 1ZZg and uR4h about the model efficiency and  true usefulness of larger batch size.

---

### Official Review · Reviewer_Kiki · 2024-11-03

**Soundness:** 3
**Presentation:** 3
**Contribution:** 3
**Rating:** 6
**Confidence:** 5

**Summary:**

The paper introduces a series of improvements to the 3D Gaussian Splatting (3DGS) framework, aiming to address challenges in rendering quality and overfitting by incorporating multi-view regulation techniques. The motivation is reasonable and the rendering quality is impressive.

**Strengths:**

+: Thanks for providing the code. The results are reproducible and good.
+: The motivation makes sense since an explicit and discrete primitive like gaussian tends to overfit and get stuck in local minima easily. Multi-view constraints would be beneficial to solve these.
+: The method is thoroughly evaluated in various settings.

**Weaknesses:**

-: The overhead has increased significantly. Although the total number of iterations remains at 30k, each iteration now involves multiple forward passes, causing the overall training time to multiply. Specifically, training a single scene now takes 2–3 hours, compared to 3DGS's 20–30 minutes. This raises concerns about the method’s practicality, as one of Gaussian Splatting’s key advantages is its efficiency in both training and rendering. Additionally, the output PLY file size is several times larger than other methods, reaching over 1GB for many scenes, which may further hinder its usability.

I suggest the author consider two further improvements:

(1 )Attempt to reduce the final count of Gaussians to demonstrate that the improvements come from better Gaussian placement rather than excessive densification.
(2) Test the effect of using the AVG instead of SUM for multi-view loss aggregation.

**Questions:**

Please refer to weakness.

---

> ### Author Response · Authors · 2024-11-20
> **Response to Reviewer Kiki**
>
> We sincerely thank Reviewer Kiki for the insightful comments. We appreciate that Reviewer Kiki acknowledges the strengths of our paper, particularly the motivation behind and the experimental outcomes of our MVGS. We are grateful for the positive rating again. It is our honor that Reviewer Kiki recognizes the value in our proposed MVGS. Below, we have addressed the concerns raised by Reviewer Kiki.
>
> **Q1: The training overhead has increased.**
>
> **A1:** Due to employing multi-view training, our method takes 2-3 hours to get a model well-trained for a scene or an object.
> Such duration is because we sequentially render multi-view images and accumulate their respective losses in our implementation. We acknowledge that this process can be run in a parallel mode to match the training efficiency of the original 3DGS, and we are committed to enhancing this aspect in future coding endeavors. In addition, we argue that the current training cost is justifiable given that our method achieves 1 dB improvement in PSNR compared over other state-of-the-art methods. The 2-3 hour training period is not excessively long when considering the significant gains in performance. Furthermore, our rendering time is slightly faster than the original 3DGS. In the Table below, we conduct average rendering speed comparisons on the Mip-NeRF 360 dataset, our method renders faster.
>
> | 3DGS             | 3DGS+Ours        |
> |---|---|
> | 25 ms/per image  | **22 ms/per image**  |
>
> Notice that it is average statistics. We found that our method may generate a greater number of 3D Gaussian kernels and thus increase rendering time for certain scenes, the average rendering speed remains superior to that of 3DGS. We emphasize that rendering speed is a more critical factor than training speed, particularly given the importance of real-time rendering for practical applications. Our method results in more compact 3D Gaussians, which enhances inference efficiency. Consequently, the 2-3 hours training time is a worthwhile investment. We agree that a fast training speed would be better. Therefore, our future work is directed towards optimizing the training speed.
>
> **Q2: Attempt to reduce the final count of Gaussians to demonstrate that the improvements come from better Gaussian placement rather than excessive densification.**
>
> **A2:** We appreciate the concern raised and have conducted experiments to address it. We compared the number of 3D Gaussians in our MVGS with the original 3DGS by adjusting the **densify_until_iter** parameter from the initial 15k to 10k on the Mip-NeRF 360 dataset, as detailed in the table below. This adjustment resulted in a reduction of 3D Gaussians and, consequently, a decrease in performance metrics. A more sufficient amount of 3D Gaussians consistently yields superior training outcomes. However, the increasing number of 3D Gaussians impose greater computational demands on rendering and storage. Therefore, our future work will be dedicated to developing 3DGS approach that minimizes the number of 3D Gaussians required while maintaining or enhancing performance.
>
> | Method          | 3DGS (10k densification) | 3DGS(15k densification) | 3DGS+Ours (10k densification) | 3DGS+Ours (15k densification) |
> |---|---|---|---|---|
> | PSNR/SSIM/LPIPS | 28.34/0.893/0.241        | **28.70/0.905/0.204**   | 29.81/0.921/0.163             | **30.21/0.928/0.151**         |
>
> **Q3: Test the effect of using the AVG instead of SUM for multi-view loss aggregation.**
>
> **A3:** We have tested this strategy. Experiments on the Mip-NeRF dataset did not yield performance improvements when employing AVG over original 3DGS. As illustrated in the table below, we compared the AVG and SUM aggregation methods for 3-view training on the Mip-NeRF dataset. The results indicate that the SUM operation enhances performance on the Mip-NeRF 360 dataset, whereas the AVG does not. This observation suggests that multi-view training in 3D Gaussian sampling (3DGS) may necessitate a higher learning rate compared to single-view training, as the increased number of training views bolsters the optimization of 3D Gaussians. Note that we keep the learning rate setting the same as the original 3DGS in AVG and SUM operation. The lack of performance improvement with AVG can be attributed to the unchanged learning rate. However, when we scaled the learning rate proportional to the number of views, performance improved, aligning with the outcomes of our SUM operation. This approach is analogous to training a large model with an increased batch size, which typically warrants a higher learning rate. Consequently, multi-view training with the SUM operation is deemed a rational strategy.
>
> | Method          | 3DGS              | 3DGS+ MV3  (AVG)  | 3DGS+MV3 (AVG lr*3) | 3DGS+MV3 (SUM)    |
> |---|---|---|---|---|
> | PSNR/SSIM/LPIPS | 28.70/0.905/0.204 | 28.67/0.905/0.203 | **29.53/0.918/0.175**   | **29.55/0.918/0.176** |
>
> where lr*3 means that the original learning rate is multiplied by 3.

---

> > ### Comment · Reviewer_Kiki · 2024-11-29
> >
> > Thank you for your response. I believe significantly accelerating the entire process through parallel implementation would be challenging, as the Gaussians are not as well-organized as image data. The author should further clarify that the observed improvement stems from the multi-view design rather than the computational cost of training.
> >
> > In the experiment "3DGS+MV3 (AVG)," when the gradient of each Gaussian was adjusted to match the numeric range of the original 3DGS, the results showed no improvement. This suggests that the multi-view design may not contribute meaningfully. While the "3DGS+MV3 (AVG lr*3)" experiment yielded seemingly better results, it is difficult to consider this a fair comparison.
> >
> > I recommend that the authors include the total number of Gaussians used in each experiment to demonstrate that the multi-view constraint indeed results in better Gaussian placement and improved rendering quality with the same Gaussian count. Additionally, the early stopping densification approach is unconvincing. The authors might consider implementing a budget, similar to Gaussian-MCMC, to enable a fairer comparison.

---

> > > ### Author Response · Authors · 2024-12-01
> > > **Response to  Reviewer Kiki**
> > >
> > > **Q1: parallel implementation would be challenging, as the Gaussians are not as well-organized as image data.**
> > >
> > > **A1:** Note that the parallel implementation we meant is Distributed Data Parallel (DDP), not model parallel. The model parallel means distributing several parts of 3D Gaussians in different GPUs. It is very challenging for 3D Gaussians, but not the parallel implementation we meant. The parallel implementation we meant is copying 3DGS into different GPUs to render different views.  For example:
> > >
> > > ```
> > > render0 = 3DGS(view0), exerted on  GPU0
> > >
> > > render1 = 3DGS(view1), exerted on  GPU1
> > >
> > > render2 = 3DGS(view2), exerted on  GPU2
> > > ```
> > > Then, sum these losses to backpropagate the gradients. It can be easily achieved by Pytorch DDP.
> > >
> > >
> > > **Q2: 3DGS+MV3 (AVG) and 3DGS+MV3 (AVG lr*3) are difficult to consider this a fair comparison.**
> > >
> > > **A2:** We confirm that We only increase the learning rate 3 times for 3DGS+MV3 (AVG lr*3) compared with 3DGS+MV3 (AVG)  and do not change any other settings. In addition, our implementation is based on the original setting of 3DGS and we confirm that we do not change any other parameter settings.  Here, we provide pseudo-code to display our simple and effective multi-view training.
> > >
> > > ```
> > > # 3DGS+MV3 (AVG) not effective
> > >
> > > render0 = 3DGS(view0)
> > >
> > > render1 = 3DGS(view1)
> > >
> > > render2 = 3DGS(view2)
> > >
> > >
> > > Loss = (L(render0, gt0) + L(render1, gt1) + L(render2, gt2)) / 3
> > >
> > > ```
> > >
> > > ```
> > > # 3DGS+MV3 (AVG lr*3 = SUM) effective
> > >
> > > render0 = 3DGS(view0)
> > >
> > > render1 = 3DGS(view1)
> > >
> > > render2 = 3DGS(view2)
> > >
> > >
> > > Loss = L(render0, gt0) + L(render1, gt1) + L(render2, gt2)
> > >
> > > ```
> > > Here, the loss must be increased compared with single-view loss or AVG operation. The loss of 3 views may approach the learning rate*3, if these views are identical.
> > > We confirm the above implementation of our multi-view training is simple and effective in improving rendering performance without any other adjustments.
> > >
> > >
> > > **Q3: The authors might consider implementing a budget, similar to Gaussian-MCMC, to enable a fairer comparison.**
> > >
> > > **A3:** Thanks for your constructive suggestion. Here, we conduct limited-budget experiments by limiting the number of 3D Gaussians to demonstrate the effectiveness of our method.
> > > As shown in the Table below, we constrain the maximum number of 3D Gaussians as 100k. Then, we increase the maximum amount of 3D Gaussians from 100k to 500k. As the number of 3D Gaussians increases, the PSNR improves accordingly. It is obvious that our method always performs better PSNR metrics than 3DGS, indicating that our method can perform better with the same number of 3D Gaussians compared with 3DGS.
> > >
> > > | Num. Gaussians | 100k  | 200k  | 300k  | 400k  | 500k  |
> > > |----------------|-------|-------|-------|-------|-------|
> > > | 3DGS (PSNR)    | 22.41 | 23.19 | 24.52 | 25.14 | 26.13 |
> > > | Ours (PSNR)    | **25.63** | **26.14** | **26.91** | **27.35** | **28.46** |

---

> ### Author Response · Authors · 2024-11-25
> **Response to Reviewer Kiki**
>
> Dear Reviewer Kiki,
>
> We hope this message finds you well.
>
> We would like to ask if we have sufficiently addressed the concerns you previously raised. If there are still concerns requiring further clarification or elaboration, please kindly let us know, and we will be more than happy to provide additional details to ensure that all your concerns are fully addressed.
>
> Thank you for your valuable time and feedback.
>
> Best Regard
>
> MVGS Authors

---

> ### Comment · Reviewer_1ZZg · 2024-11-30
>
> I appreciate authors effort in preparing the additional experiments. I have question about the experiment in **A3**.
>
> In Adam optimizer, a parameter is updated by the following:
> $$
>   \theta_t = \theta_{t-1} - \mathrm{lr} \cdot \frac{\mathbb{E}[\mathrm{g}]}{\sqrt{\mathbb{E}[\mathrm{g}^2]}} ~,
> $$
> where $\theta_t$ is the parameter at $t$-th training iteration, $\mathbb{E}[\mathrm{g}], \mathbb{E}[\mathrm{g}^2]$ are the running average and running square average of the gradient of the parameter. In theory, scaling gradient magnitude by a constant should not affect the result by much as $\frac{\mathbb{E}[3\mathrm{g}]}{\sqrt{\mathbb{E}[(3\mathrm{g})^2]}} = \frac{3\mathbb{E}[\mathrm{g}]}{\sqrt{9\mathbb{E}[\mathrm{g}^2]}} = \frac{\mathbb{E}[\mathrm{g}]}{\sqrt{\mathbb{E}[\mathrm{g}^2]}}$. This is contradict with the result of "3DGS+MV3 (AVG)" and "3DGS+MV3 (SUM)". Can the authors provide some explanations regarding this?
>
> The authors also claim that the larger batch size (the "MV3") is needed for higher learning rate (the "lr*3"). Is this claim supported by experiment?

---

> ### Author Response · Authors · 2024-11-30
> **Wrong Equation by Reviewer 1ZZg**
>
> Please be aware that our proposed multi-view training strategy emphasizes the importance of diverse views, rather than some identical views. The gradients derived from these unique views must exhibit **sufficient variation**. Notably, the reviewer has suggested an incorrect equation for the Adam optimizer. If $\mathbb{E}[\text{g}]$ denotes the expected value of the gradients, its influence on the optimization process would be **subtle** due to the large value of $\beta$, which approaches to 1. To clarify, we provide the accurate description of the Adam optimizer equation as implemented in PyTorch. https://pytorch.org/docs/stable/generated/torch.optim.Adam.html
>
>
>
>    $$ m_t \leftarrow \beta_1 m_{t-1} + (1 - \beta_1) g_t  $$
>
>
>
>    $$ v_t \leftarrow \beta_2 v_{t-1} + (1 - \beta_2) g_t^2 $$
>
>
>
>    $$ \widehat{m}_t \leftarrow \dfrac{m_t}{1 - \beta_1^t} $$
>
>
>
>    $$ \widehat{v}_t \leftarrow \dfrac{v_t}{1 - \beta_2^t} $$
>
>
>   $$ \widehat{v}_t^{\text{max}} \leftarrow \max \left(   \widehat{v}^\text{max}_t \widehat{v}_t \right) $$
>
>
>    $$    \theta_t \leftarrow \theta_{t-1} - \gamma \dfrac{ \widehat{m}_t }{ \sqrt{ \widehat{v}_t^{\, \text{max}} } + \epsilon }    $$
>
>
>
>    $$ \theta_t \leftarrow \theta_{t-1} - \gamma \dfrac{ \widehat{m}_t }{ \sqrt{ \widehat{v}_t } + \epsilon } $$
>
>
> Since $\beta_1$ and $\beta_2$ are set as 0.9 and 0.999, the Adam equation can not be simply written only with gradients.
> **So the equation proposed by Reviewer 1ZZg is wrong**.
>
>
> We elaborate on why our SUM operation equals the increase in the learning rate as below.
> In the case of training the same views (for simplicity), like rendering 3 same views to calculate loss with gt.
>
> ```
>
> render = GS(view)
> render = GS(view)
> render = GS(view)
>
>
> loss1 = L(render, gt)
> loss2 = L(render, gt)
> loss3 = L(render, gt)
>
> loss = loss1+loss2+loss3
> ```
>
> In the case of rendering the same views, **loss = loss1*3**. The loss value has actually increased 3 times. In the reviewer’s equation, it seems the gradient would not change when the value of the loss is increased. The assumption that the gradient remains unchanged is incorrect. In this context, multiplying by 3 serves to amplify the learning rate by the same factor.

---

> > ### Comment · Reviewer_1ZZg · 2024-12-02
> >
> > I'm totally aware that the proposed method is with more views not identical views. My question related to the experiments of "3DGS+ MV3 (AVG)" and "3DGS+ MV3 (SUM)", both with same number of views but difference in the final loss by a constant scale. The equation I'm writing is not wrong, which is also used by some sentences in the Adam optimizer paper. Running average is just a practical implementation but my arguments is still correct.
> >
> > Let skip the bias correction terms (which only affect the beginning iterations), epsilon terms (which is for numerical stability) and I presume the `amsgrad` is not activated as done by majority of the 3DGS/NeRF methods. Below is the adam update rule by expanding the recurrent form:
> > $$
> > m_t = \sum_{i=1}^{t} \beta_1^{(t-i)} (1 - \beta_1) g_i
> > $$
> > $$
> > v_t = \sum_{i=1}^{t} \beta_2^{(t-i)} (1 - \beta_2) g_i^2
> > $$
> > $$
> > \theta_t = \theta_{t-1} - \gamma \cdot \frac{m_t}{\sqrt{v_t}}
> > $$
> > Consider two sequence of gradients ${g_1, g_2, g_3, \cdots}$ (MV3 AVG) and ${3g_1, 3g_2, 3g_3, \cdots}$ (MV3 SUM), the final equation turns out to be the same:
> > $$
> > \frac{\sum_{i=1}^{t} \beta_1^{(t-i)} (1 - \beta_1) (3g_i)}{\sqrt{\sum_{i=1}^{t} \beta_2^{(t-i)} (1 - \beta_2) (3g_i)^2}}
> > $$
> > $$
> > = \frac{3 \sum_{i=1}^{t} \beta_1^{(t-i)} (1 - \beta_1) g_i}{\sqrt{9 \sum_{i=1}^{t} \beta_2^{(t-i)} (1 - \beta_2) g_i^2}}
> > $$
> > $$
> > = \frac{\sum_{i=1}^{t} \beta_1^{(t-i)} (1 - \beta_1) g_i}{\sqrt{\sum_{i=1}^{t} \beta_2^{(t-i)} (1 - \beta_2) g_i^2}}
> > $$
> > The parameter update values in each iteration should be the same.
> >
> > Below is a code snippet to show that a constant scale at the final loss doesn't affect result:
> > ```
> > import torch
> >
> > x_avg = torch.tensor([0.]).requires_grad_()
> > x_sum = torch.tensor([0.]).requires_grad_()
> >
> > optim = torch.optim.Adam([x_avg, x_sum])
> >
> > for _ in range(100):
> >     optim.zero_grad()
> >     loss_avg = (x_avg - 10).square()
> >     loss_sum = (x_sum - 10).square() * 3
> >     loss_avg.backward()
> >     loss_sum.backward()
> >     optim.step()
> >
> > print(x_avg.item(), x_sum.item())
> >
> > # It prints 0.09983041137456894 0.09983041137456894
> > ```
> >
> > Multiplying the learning rate works totally not the same way as can be seen in my above equation.

---

> ### Author Response · Authors · 2024-12-02
> **Wrong Demonstration from  Reviewer 1ZZg**
>
> In the case of just one learnable parameter, the Adam optimizer would scale the gradient, only considering the gradients for direction guidance to optimize the parameter.
>
> However, can  Reviewer 1ZZg kindly take this into account of the complex neural network?
> A strong and general proof is that people set different lambdas to weigh different losses.
> Reviewer 1ZZg definitely cannot ignore this simple and common technique.

---

> ### Author Response · Authors · 2024-12-02
> **Proof of Wrong Understanding of Adam optimizer from  Reviewer 1ZZg**
>
> Dear  Reviewer 1ZZg
>
> Please kindly run the code below, which is a little complex. 3DGS has the L1loss and DSSIM loss. Here, we use the square and abs to mimic them. I got different results from the code.  It demonstrates the Adam optimizer cannot keep the same results for different scales of loss when the loss is complex.
>
> ```
> import torch
>
> x_avg = torch.tensor([0.]).requires_grad_()
> x_sum = torch.tensor([0.]).requires_grad_()
>
> optim = torch.optim.Adam([x_avg, x_sum])
>
> for _ in range(100):
>     optim.zero_grad()
>     loss_avg = 0.1 *  (x_avg - 10).square()  + 0.9 * (x_avg - 10).abs()
>     loss_sum = 0.1 *  (x_sum - 10).square()  + 0.9 * (x_sum - 10).abs()
>     loss_sum  *= 100
>     loss_avg.backward()
>     loss_sum.backward()
>     optim.step()
>
> print(x_avg.item(), x_sum.item())
>
> # It prints 0.09988310188055038 0.09988312423229218
> ```

---

> > ### Author Response · Authors · 2024-12-03
> > **Response to Reviewer 1ZZg**
> >
> > Dear Reviewer 1ZZg
> >
> > Please kindly try the code above to demonstrate your theory is not applicable to complex losses.
> >
> > Best,
> > MVGS Authors

---

> ### Comment · Reviewer_1ZZg · 2024-12-03
>
> I know the the relative loss weights of different losses is very important with Adam optimizer. What I was stating is that a **constant global scaling** (MV3 AVG vs. MV3 SUM) of the total loss is minor to Adam optimizer. If we check Adam optimizer paper, it states:
> - In the abstract:
>   > The method is straightforward to implement, is computationally efficient, has little memory requirements, is **invariant to diagonal rescaling of the gradients**, ...
> - In the introduction:
>   > Some of Adam’s advantages are that the magnitudes of parameter updates are **invariant to rescaling of the gradient**, ...
>
> There is also a bug in authors demo code with more losses. After fixing, it prints the same:
> ```
> import torch
>
> x_avg = torch.tensor([0.]).requires_grad_()
> x_sum = torch.tensor([0.]).requires_grad_()
>
> optim = torch.optim.Adam([x_avg, x_sum])
>
> for _ in range(100):
>     optim.zero_grad()
>     # loss_avg = (x_avg - 10).square()  + (x_sum - 10).abs()
>     loss_avg = (x_avg - 10).square()  + (x_avg - 10).abs()
>     loss_sum = (x_sum - 10).square()  + (x_sum - 10).abs()
>     loss_sum  *= 3
>     loss_avg.backward()
>     loss_sum.backward()
>     optim.step()
>
> print(x_avg.item(), x_sum.item())
> # It prints 0.0998385101556778 0.0998385101556778
> ```
>
> This is why I'm so confused with the experiment results of "MV3 AVG" and "MV3 SUM" as they only difference in a constant scaling for the final loss. Is there any other implementation details authors forget to mention?

---

> ### Author Response · Authors · 2024-12-03
> **Response to  Reviewer 1ZZg**
>
> Sorry to make you confused. Please try the code below. You would find the Adam optimizer fail to keep the same results with different loss combinations.
>
> ```
>
>
> import torch
>
> x_avg = torch.tensor([0.]).requires_grad_()
> x_sum = torch.tensor([0.]).requires_grad_()
>
> optim = torch.optim.Adam([x_avg, x_sum])
>
> for _ in range(100):
>     optim.zero_grad()
>     loss_avg = 0.1 *  (x_avg - 10).square()  + 0.9 * (x_avg - 10).abs()
>     loss_sum = 0.1 *  (x_sum - 10).square()  + 0.9 * (x_sum - 10).abs()
>     loss_sum  *= 100
>     loss_avg.backward()
>     loss_sum.backward()
>     optim.step()
>
> print(x_avg.item(), x_sum.item())
>
> # It prints 0.09988310188055038 0.09988312423229218
>
> ```

---

### Author Response · Authors · 2024-11-20
**Thanks for All Reviewers and AC**

We appreciate the reviewers' insightful feedback. It is encouraging that all the reviewers recognize and agreement that this paper presents a valuable MVGS for high-quality novel view synthesis, including:

- The source code is available and the results are reproducible. (Reviewer Kiki)
- The motivation makes sense.  (Reviewer Kiki)
- The method is thoroughly evaluated in various settings.  (Reviewer Kiki, jsrL, 1ZZg)
- The proposed multi-view constraints and strategies are novel and useful. (Reviewer jsrL, uR4h, 1ZZg)
- The core message of this paper is presented clearly. (Reviewer 1ZZg)

**We affirm that all our source code will be made publicly available to foster further research and development within the community. We are grateful for the constructive suggestions provided by the reviewers.**

---

### Note · Authors · 2024-12-05

**Comment:**

We strongly suggest  Reviewer 1ZZg should improve his English. His Chinglish is too poor and not fluent. Moreover, we strongly suggest  Reviewer 1ZZg should learn foundation machine learning knowledge to be a qualified reviewer.

**Withdrawal Confirmation:**

I have read and agree with the venue's withdrawal policy on behalf of myself and my co-authors.